# Topological Inductive Bias fosters Multiple Instance Learning in Data-Scarce Scenarios

**Salome Kazeminia**      *salome.kazeminia@helmholtz-munich.de*
*Helmholtz Munich*
*Technical University of Munich*

**Carsten Marr**      *carsten.marr@helmholtz-munich.de*
*Helmholtz Munich*
*Ludwig-Maximilian-University Hospital*
*University of Munich*
*DKTK, German Cancer Consortium*
*Munich Center for Machine Learning*

**Bastian Rieck**      *bastian.grossenbacher@unifr.ch*
*Helmholtz Munich*
*University of Fribourg*

**Reviewed on OpenReview:** *https://openreview.net/forum?id=1hZy9ZjjCc*

## Abstract

Multiple instance learning (MIL) is a framework for weakly supervised classification, where labels are assigned to sets of instances, i.e., bags, rather than to individual data points. This paradigm has proven effective in tasks where fine-grained annotations are unavailable or costly to obtain. However, the effectiveness of MIL drops sharply when training data are scarce, such as for rare disease classification. To address this challenge, we propose incorporating topological inductive biases into the data representation space within the MIL framework. This bias introduces a topology-preserving constraint that encourages the instance encoder to maintain the topological structure of the instance distribution within each bag when mapping them to MIL latent space. As a result, our Topology Guided MIL (TG-MIL) method enhances the performance and generalizability of MIL classifiers across different aggregation functions, especially under scarce-data regimes. Our evaluations show average performance improvements of 15.3% for synthetic MIL datasets, 2.8% for MIL benchmarks, and 5.5% for rare anemia classification compared to current state-of-the-art MIL models, where only 17–120 samples per class are available. We make our code publicly available at `https://github.com/SalomeKaze/TGMIL`.

## 1 Introduction

Multiple Instance Learning (MIL) is a variant of weakly supervised learning that operates without annotations for individual data points. In MIL, each 'bag,' which represents a group of instances, is assigned a single label (Babenko, 2008; Lu et al., 2020). A bag is labeled positive if it contains at least one positive instance; otherwise, it is labeled negative. For example, in blood sample–based disease classification, each sample can be viewed as a bag of individual cells. The sample is labeled positive if it contains any atypical cells and negative otherwise.

Proper instance representation is critical for ensuring model reliability in clinical decision-making. However, data scarcity, a common issue in the diagnosis of rare diseases, hinders MIL models' ability to learn such representations. In such cases, the need for MIL-based training approaches that operate in the scarce-data regime is paramount.

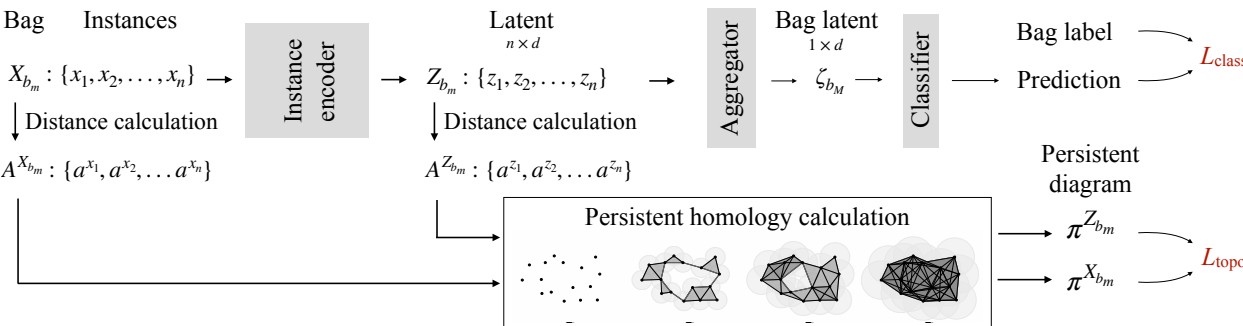

Figure 1: Topologically Guided Multiple Instance Learning (TG-MIL): We calculate the distance matrix $A$ of input instances $x_i$ inside each bag $X_{b_m}$. Subsequently, we apply persistent homology based on the Vietoris-Rips complex by treating each bag's instances as a point cloud. We employ the same process for the latent feature vectors $Z$ of each bag. Generating shape descriptors (*persistence diagrams* $\pi$) for both the latent space and the image space representations of the bag, we calculate a topological loss ($L_{\text{topo}}$) and combine it with the standard MIL loss ($L_{\text{class}}$).

A potential solution is to leverage additional structure from data via inductive biases, a strategy that has shown promise in other machine learning domains (Goyal & Bengio, 2022). We treat each bag as a point cloud in a high-dimensional data space, consistent with Deep Sets and Pointnet (Zaheer et al., 2017; Qi et al., 2017), and define an inductive bias based on the bag's topological features. These features should be preserved after embedding bag instances into the model's latent space, ensuring that the data's essential topological properties are maintained. Being able to capture fundamental topological principles of data at multiple scales, topological algorithms[1] recently arose as a source of such inductive biases, permitting the integration into deep learning models (Hensel et al., 2021). The primary appeal of such algorithms lies in their robustness to noise and perturbations, resulting in *stable multi-scale representations*. Also, these algorithms improve generalizability and predictive performance when a pronounced topological signal is present in the data (Horn et al., 2022; Waibel et al., 2022), even in the presence of *singular structures*, which preclude the use of standard techniques (von Rohrscheidt & Rieck, 2023).

We introduce *Topologically Guided Multiple Instance Learning (TG-MIL)*, a data-centered solution to address the challenges of training MIL with scarce data in an end-to-end manner. By leveraging multi-scale shape descriptors on the level of MIL bags, we develop a scheme that preserves crucial topological information in the latent space of our model (see Figure 1 for a schematic overview). We demonstrate that maintaining even only the connectivity-based topological bias (captured via 0D persistent homology) inherent in a bag's data distribution improves the performance of MIL classifiers. This topological bias, reflecting each bag's natural structure and relationships, carries critical information for accurate classification. Preserving this intrinsic structure allows MIL classifiers to learn and generalize more effectively, improving performance accuracy, robustness, and adaptability across different training data. The **main contributions** of our work are:

- We develop TG-MIL, the first topological method to improve the generalizability of MIL to data-scarce scenarios.
- Our method can be integrated with any MIL aggregation strategy, improving performance under data scarcity in an end-to-end training setting.
- TG-MIL outperforms the state-of-the-art on MIL benchmarks and rare anemia classification.

## 2 Background

**MIL Architectures.** MIL architectures typically comprise three key components: an instance encoder, an aggregation function, and a classifier (Figure 1). Given a collection of $M$ bags $b_1, \ldots, b_M$, each bag $b_m$ contains a set of instances, represented as $X_{b_m} := \{x_1, \ldots x_n\}$ with $n$ denoting the number of instances in the

---

[1]Despite their name, these algorithms also capture geometrical aspects of data, but we will refrain from writing *geometrical-topological algorithms* for brevity.

bag. An instance encoder $f_\theta$ with parameters $\theta$ transfers each instance into a latent space, yielding feature vectors $z_i := f_\theta(x_i)$. The aggregation function then creates a global representation of the bag $\zeta_{b_m}$ from these embedded instances, from which the classifier head predicts the bag's overall label.

**Geometry & Topology.** Our work is based on advances in topological machine learning (Hensel et al., 2021), a nascent field that aims to leverage the geometry and topology of data to elicit improved representations. We employ *persistent homology*, a technique for calculating multi-scale topological information from data (Edelsbrunner & Harer, 2009). Persistent homology treats data as a point cloud and aims to define this cloud using simple geometric elements (simplices) that describe the overall geometry of the data, such as connected components, loops, higher-dimensional voids, and geometric properties like curvature or convexity (Bubenik et al., 2020; Turkes et al., 2022). This information is collected in a set of *persistence diagrams*, which serve as multi-scale topological descriptors. These descriptors are calculated by approximating the data with a simplicial complex—a generalized graph—typically based on distance functions like the Euclidean distance (Figure 1). Recent work proved that persistent homology can be integrated with deep learning models, leading to a new class of hybrid models that are capable of capturing topological aspects of data. Such models have shown exceptional performance as regularization terms in different applications (Chen et al., 2019; Vandaele et al., 2022; Waibel et al., 2022).

## 3 Related Work

Recent advances in MIL can be categorized into two main methods and use cases: i) Methods using *transfer learning* for feature extraction and only focusing on training aggregation mechanisms (Shao et al., 2021; Zhang et al., 2022; Liu et al., 2023; Tang et al., 2023; Fourkioti et al., 2024; Castro-Macías et al., 2024). These approaches are designed for histopathology applications and have not been evaluated on basic MIL problem settings. Bags typically consist of thousands of spatially correlated patch instances extracted from a single whole-slide image. Due to architectural and computational constraints, instance representations are kept fixed using frozen pre-trained encoders, and end-to-end training is not supported. In contrast, our work focuses on improving instance representation learning in end-to-end MIL settings with spatially independent instances and limited training data. Adapting the above methods would require removing certain components (e.g., positional embeddings in TransMIL (Shao et al., 2021)), reducing the complexity of the aggregation model, and considering a proper learnable instance encoder. These changes would substantially alter the original architectures and make a fair comparison non-trivial. Therefore, we exclude these approaches from aggregation comparisons, while noting that our proposed framework is, in principle, compatible with such aggregation mechanisms. ii) *End-to-end* approaches that optimize both bag aggregation and instance representation during training. Ilse et al. (2018) introduced the Attention-based MIL (APMIL), which learns instance-level attention weights, and the Gated Attention MIL (GAPMIL) variant, which incorporates a gating mechanism to modulate those weights. Recently, APMILwD and GAPMILwD (Zhu et al., 2025) extended the classical attention-based frameworks by introducing instance-level dropout, which regularizes attention learning and significantly improves generalization, achieving state-of-the-art results on classic MIL benchmarks. However, the attention-based pooling mechanism (see Appendix, equations 13 and 14) may still lack reliability, as it does not uniformly enhance representation across all instances, while, in medical diagnosis and drug discovery, accurate identification of individual components (e.g., cells in microscopic blood sample images or chemical compounds) is crucial for making accurate predictions at the sample level. Other end-to-end methods have focused on designing alternative aggregation mechanisms to better capture complex instance relationships and improve MIL classification. BDRMIL (Huang et al., 2022) modeled pairwise relations between instances within a bag, capturing inter-instance dependencies. DistNet (Oner et al., 2023) proposed a distribution-aware formulation that represents each bag via statistical properties of its instance embeddings, enabling aggregation through learned distributional representations rather than attention scores. RGMIL (Du et al., 2023) further enhanced instance-level representation through a regressor-guided aggregator that aligns instance- and bag-level predictions, improving the signal flow through the encoder. Although this approach refines instance-level representation and enhances overall MIL performance, it struggles to accurately capture nuanced variations in the data when training data is scarce.

In medical applications, like classifying rare anemia disorders, key factors such as the severity of cell deformation and the ratio of deformed cells in a blood sample are paramount. Thus, it is essential that instance-level representations capture the distribution of cell classes while maintaining an interpretable bias toward deformation severity, thereby ensuring both the reliability and interpretability of the MIL model's predictions. State-of-the-art approach (Kazeminia et al., 2022) in this domain refines aggregation techniques, i.e., anomaly-aware pooling, to encourage the encoder to push healthy cells toward a normal distribution. However, optimal performance of this method still relies on a large amount of training data, thus limiting its potential to achieve higher accuracy in data-scarce environments.

To overcome the challenges posed by data scarcity, we introduce a topological inductive bias that incentivizes models to preserve topological information in latent representations. Our method thus falls into the second category of end-to-end techniques and emphasizes the crucial role of data representations in MIL.

## 4 Methods

We consider each bag as a point cloud in a high-dimensional space whose topological features should be adequately captured by the model. Each instance influences the bag's "shape," with positive instances notably altering its shape in comparison to the distribution of negative samples. We thus need a descriptor that captures the characteristics of a point cloud while remaining *stable* to perturbations and invariant under transformations irrelevant to determining the overall shape, such as translations and rotations. Persistent homology provides a suitable descriptor, as Sheehy (2014) demonstrates that critical topological-geometrical features captured by persistent homology are approximately preserved even under projections or embeddings of the data, making it highly robust.

The calculation of persistent homology requires a choice of distance metric. While our framework remains agnostic to the specific choice of distance metric, we have chosen to use the per-pixel Euclidean distance for its ease of calculation and empirical evidence from Moor et al. (2020a), which demonstrates the adequacy of Euclidean distances in capturing topological features. We calculate the Vietoris–Rips simplicial complex $(\mathfrak{R}_\epsilon(X_{b_m}))$ at each distance $\epsilon$, which yields all subsets of $X_{b_m}$ such that the pairwise distance between instances $x_i$ and $x_j$ is less than or equal to $\epsilon$.

$$\mathfrak{R}_\epsilon(X_{b_m}) := \{x_i | \exists x_j \in X, x_i \neq x_j : \|x_i - x_j\|_2 \leq \epsilon\} \tag{1}$$

Formally, this leads to a filtration of simplicial complexes $\{\mathfrak{R}_{\epsilon_0}(X_{b_m}), \mathfrak{R}_{\epsilon_1}(X_{b_m}), \ldots, \mathfrak{R}_{\epsilon_m}(X_{b_m})\}$, with an ordered sequence of distance thresholds $0 = \epsilon_0 < \epsilon_1 < \ldots < \epsilon_m$ (Figure 1). Persistent homology tracks the evolution of topological features (e.g., connected components (0D), loops (1D), voids (2D), etc) across different distances. Consistent with previous work (Moor et al., 2020b), we restrict most of our experiments to 0D topological features, as incorporating higher-dimensional topological features substantially increases computational cost. Given the distance matrix $A$, we consider the sorted distance values as the $\epsilon$ set at each space. For each $\epsilon_i \in \epsilon$, if a topological feature is born or destroyed, $\epsilon_i$ is recorded as a persistent edge, and the pair of data points whose interaction causes this event is recorded as the persistence pair $\pi_i$.

For example, let us consider 0D topological features for two connected components of points, $\{x_1, x_2, x_3\}$ and $\{x_4, x_5, x_6\}$. Let $\epsilon_2$ be the distance between $x_1$ and $x_6$, representing the smallest distance that connects these two components. Thus, $\epsilon_2$ is the persistent edge, and $(x_1, x_6)$ is the corresponding persistent pair. The collection of persistent edges and pairs for all $\epsilon$ is called the topological signature of the point cloud. Thus, the topological signature of bag $b_m$ in input space contains $A^{X_{b_m}}$ and $\pi^{X_{b_m}}$. Similarly, $A^{Z_{b_m}}$ and $\pi^{Z_{b_m}}$ represent the topological signature of the bag in the latent space.

Our objective is to incorporate the topological signature of the input space as an inductive bias into the model. We define a loss term to penalize the encoder $f_\theta$ for any inconsistency in the bag's topological signature during projection from input to latent space. To this end, we utilize the topological loss proposed by Moor et al. (2020b), which enables backpropagation through topological descriptors. This loss penalizes differences of topological signature of the latent space with the input space using an $\|\cdot\|_2$-norm. Specifically, first, the persistent edges in the two input and latent spaces, using the input persistence pairs, are compared using $L_{X_{b_m} \to Z_{b_m}}$. Then, the persistent edges in the two spaces using the latent persistence pairs are compared

$L_{Z_{b_m} \to X_{b_m}}$. We account for both losses to enforce consistency between the two mappings (data → latent and latent → data), while remaining invariant to the ordering of instances within each bag. The final topological loss is defined as

$$L_{\text{topo}} := L_{X_{b_m} \to Z_{b_m}} + L_{Z_{b_m} \to X_{b_m}}, \tag{2}$$

where

$$L_{X_{b_m} \to Z_{b_m}} := \frac{1}{2} \left\| A^{X_{b_m}} \left[ \pi^{X_{b_m}} \right] - A^{Z_{b_m}} \left[ \pi^{X_{b_m}} \right] \right\|^2, \tag{3}$$

and

$$L_{Z_{b_m} \to X_{b_m}} := \frac{1}{2} \left\| A^{Z_{b_m}} \left[ \pi^{Z_{b_m}} \right] - A^{X_{b_m}} \left[ \pi^{Z_{b_m}} \right] \right\|^2. \tag{4}$$

The topological loss is defined in terms of pairwise distances between instances within a bag. It makes any permutation of instances to change only their ordering and leaves the set of pairwise distances unchanged. Under the assumption that pairwise distances are unique, their relative ordering is also preserved. Consequently, the induced topological loss is invariant to arbitrary permutations of instances.

Our framework is flexible and can integrate any aggregation function to represent the whole bag $\zeta_{b_M}$. Similar to standard MIL models, we train the MIL classification head using cross-entropy loss. Our formulation also yields a variant of a multi-classifier head approach, such as the auxiliary loss proposed by Kazeminia et al. (2022). The final loss of our topologically guided MIL (TG-MIL) framework, $L_{\text{total}}$, is the weighted sum of the MIL classification loss $L_{\text{class}}$ and topological term $L_{\text{topo}}$:

$$L_{\text{total}} = L_{\text{class}} + \lambda L_{\text{topo}}, \tag{5}$$

where $\lambda$ is a hyperparameter to adjust the influence of the topological loss.

**Complexity and Parameters.** The computational complexity involved in calculating certain topological features is comparable to the rate of the inverse Ackermann function (Cormen et al., 2022), which increases significantly slower compared to the rate of increasing $n$. The computational complexity of the topological signature calculation of a bag containing $n$ instances is dominated by the calculation of pairwise distances, i.e., $\mathcal{O}(n^2)$, considering that only 0D topological features are used. The topological signature calculation does not introduce any additional learnable parameters, thereby keeping the model's parameter size unchanged. It introduces only one topological loss and one hyperparameter, denoted by $\lambda$.

**Limitations.** The primary limitation of our approach is that the calculation of topological features does *not* exhibit favorable scaling parameters in case that higher-order topological features are required. While our implementation supports topological features of arbitrary dimension, their calculation scales progressively worse; connected components, i.e., 0D features, can still be efficiently calculated (see previous paragraph), but higher-order features may prove limiting. We plan on investigating mitigation strategies in future work, using, e.g., approximate filtrations (Sheehy, 2013) or distributed computations (Wagner et al., 2021).

**Instance Learnability.** Each bag $B = \{x_1, \ldots, x_n\}$ contains instances with latent instance labels $y_i \in \{0, 1\}$ for a binary classification task. The bag label is generated by a monotone aggregation rule

$$Y_{b_M} = \phi(y_1, \ldots, y_n), \tag{6}$$

such as the standard OR assumption in MIL. Changing any instance label from negative to positive cannot change a positive bag label to a negative one. This assumption holds for the aggregation functions used in this work. As our proposed method does not introduce any new aggregation function, we omit further demonstrations for the aggregations considered. We focus on the distinction between instance-level and bag-level operations, where $g_\psi$ represents the bag-level operation, which jointly encompasses the aggregation function and the predictor.

$$\hat{Y}_{b_M} = g_\psi \left( \{ f_\theta(x_i) \}_{i=1}^n \right). \tag{7}$$

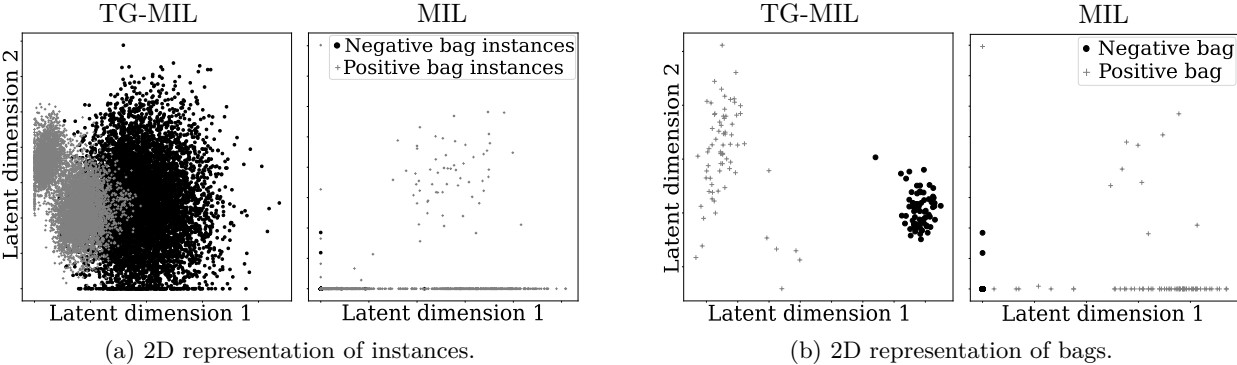

(a) 2D representation of instances.          (b) 2D representation of bags.

Figure 2: TG-MIL achieves a more distinguished representation of negative and positive bags. For a toy dataset, instances are sampled from a hypersphere when projecting them to the 2D latent space (a), leading to a more distinguished latent representation of bags (b).

Following Jang & Kwon (2024), we define the bag-level risk as

$$R_{\text{bag}} = \mathbb{E}\big[L_{class}(\hat{Y}, Y)\big] \tag{8}$$

and the instance-level risk as

$$R_{\text{inst}}(h) = \mathbb{E}\big[L_{class}(h(f(x)), y)\big], \tag{9}$$

where h(.) is an explicit instance predictor. Considering $R_{\text{inst}}(h)$ as the true instance risk, there exists an empirical instance risk $\hat{R}_{\text{inst}}(h)$ trained on the finite training set of size $m$

$$\hat{R}_{\text{inst}}(h) = \frac{1}{m} \sum_{i=1}^{m} L_{class}(h(f(x)), y). \tag{10}$$

In general $R_{\text{inst}}(h) \neq \hat{R}_{\text{inst}}(h)$ due to finite training data and model capacity. The larger this gap becomes, the poorer the instance learnability is. In our framework, the topological loss acts exclusively on the instance encoder $f_\theta$ and therefore directly influences only the instance-level risk $R_{\text{inst}}$. It does not modify the aggregation function or the bag-level predictor $g_\psi$, and hence cannot directly reduce the bag-level risk $R_{\text{bag}}$. Instead, the topological inductive bias restricts the hypothesis space of the instance encoder, improving the learnability and robustness of instance-level representations. Such improvements at the instance level can indirectly benefit bag-level inference through the shared encoder, while preserving the standard MIL aggregation assumptions. Thus, our proposed approach satisfies the necessary conditions for instance learnability without introducing additional instance-level supervision or altering the bag-level decision rule.

**Toy dataset.** To demonstrate the importance of preserving the topology of bags in MIL, we utilize a toy dataset. In this dataset, negative instances are sampled randomly from a 100D space, while positive instances are sampled from the surface of a 100D sphere—an established geometric object. To satisfy the positive bag definition of MIL, the sphere overlaps with the space of random negative instances. We consider a simple 2-layer encoder projecting instances from the 100D space to a visualizable representation (see Figure 2). We apply our framework utilizing regressor-guided aggregation (Du et al., 2023) as the baseline MIL model (see Appendix Table 5). Figure 2 illustrates the resulting instance and bag representations, contrasting the scenarios with and without topological guidance. TG-MIL preserves the topology of positive instances, resembling a circle, as the expected 2D projection of a hypersphere. As a result, the aggregated bag's latent is more distinct, leading to a higher classification accuracy (MIL and TG-MIL yield $0.55 \pm 0.05$ and $0.80 \pm 0.22$ accuracy, respectively, averaged over 5 runs).

## 5 Experiments

We evaluate TG-MIL's performance across different datasets. Synthetic MIL datasets provide a controlled environment for investigating TG-MIL's performance across varying amounts of training data. MIL bench-

marks offer insight into its general effectiveness across heterogeneous data modalities while facilitating a fair comparison with state-of-the-art methodologies. Furthermore, our evaluation contains a real-world application, i.e., the classification of *rare anemia*, that presents additional challenges inherent to the MIL problem domain, such as the contribution of severity and the ratio of positive instances within the bag class.

## 5.1 Synthetic Datasets

To evaluate the robustness of the TG-MIL framework across varied MIL data properties, including the number of training bags, instance image complexity, and bag sizes, we draw on the methods outlined by Ilse et al. (2018). We create two series of synthetic datasets: the first comprises bags of MNIST images as instances, and the second comprises bags of Fashion MNIST (Xiao et al., 2017) images a more challenging scenario with complex visual data. In constructing the MIL synthetic datasets, a bag is labeled positive if it contains at least one instance of the digit "9" in MNIST or the "Dress" class in Fashion MNIST; otherwise, it is labeled *negative*. We construct distinct training datasets containing 10, 14, 20, 50, 100, and 200 bags to evaluate the influence of the quantity of training data. Additionally, we explore different instance counts per bag, sampling from Gaussian distributions with means and standard deviations of $(10, 2)$, $(50, 10)$, and $(100, 20)$, respectively. Positive bags are defined as those containing at least one positive instance, accounting for up to 20% of the instances within the bag.

**Models.** We use a deep instance encoder architecture introduced by Ilse et al. (2018) (see Appendix Table 6). It consists of two convolutional layers with kernel size 5, stride 1, and ReLU activation. These layers generate 20 and 500 feature maps, respectively, followed by a fully-connected layer. The attention network comprises two linear layers, resulting in a final output dimension of 128 followed by 1. The topological signature of input instances is calculated in image space and latent space, using pixel-wise Euclidean distances between instance images and latent feature vectors (Figure 1).

**Results.** We evaluate the effectiveness of topological guidance utilizing three aggregation functions in MIL: max pooling, average pooling, and attention-based pooling (Ilse et al., 2018), which serves as the baseline for numerous studies in the field, in addition to the regressor-guided pooling technique (Du et al., 2023). We analyze the average F1-score and its standard deviation for different numbers of training bags (Figure 3) and bag sizes (Figure 8 in the appendix) over five runs. Without topological guidance, models trained with few training bags perform poorly, akin to random guessing, while adding topological guidance provides a reasonable complexity for the encoder to resolve overfitting (Figure 9 in the Appendix shows learning curves of MIL and TG-MIL for MNIST synthetic data) and improves the MIL model performance across both datasets. Notably, topological guidance narrows the performance gap between basic aggregations of max pooling and average pooling compared to advanced attention and regressor-guided pooling techniques.

We conduct a statistical analysis using the Wilcoxon rank-sum test (Haynes et al., 2013) to assess the significance of the improvements resulting from topological guidance (Table 1). Given the 24 tests for each dataset, we applied the Bonferroni correction (Weisstein, 2004) to account for multiple comparisons, reducing the significance level from 0.05 to approximately $(\frac{0.05}{24} \approx)$ 0.002. Topological guidance significantly enhances classification in 7 out of 48 cases, particularly when using average pooling (Table 1).

## 5.2 MIL Benchmarks

**Dataset.** We assess the performance of TG-MIL using five benchmark MIL datasets commonly employed in evaluating end-to-end MIL methods in the literature. These include three image-based datasets (FOX, TIGER, and ELEPHANT) described in Dietterich et al. (1997), each comprising 200 bags. In these datasets, only tabular features of instances are available. Although these classic benchmarks are relatively small and of limited quality, we include them to ensure a fair and consistent comparison with previous MIL approaches.

We evaluate our method on the MUSK1 and MUSK2 datasets (Andrews et al., 2002), which contain data from 92 and 102 molecules, respectively. In these datasets, each molecule is represented as a bag of instances, where each instance corresponds to a different molecular conformation. The number of instances per bag

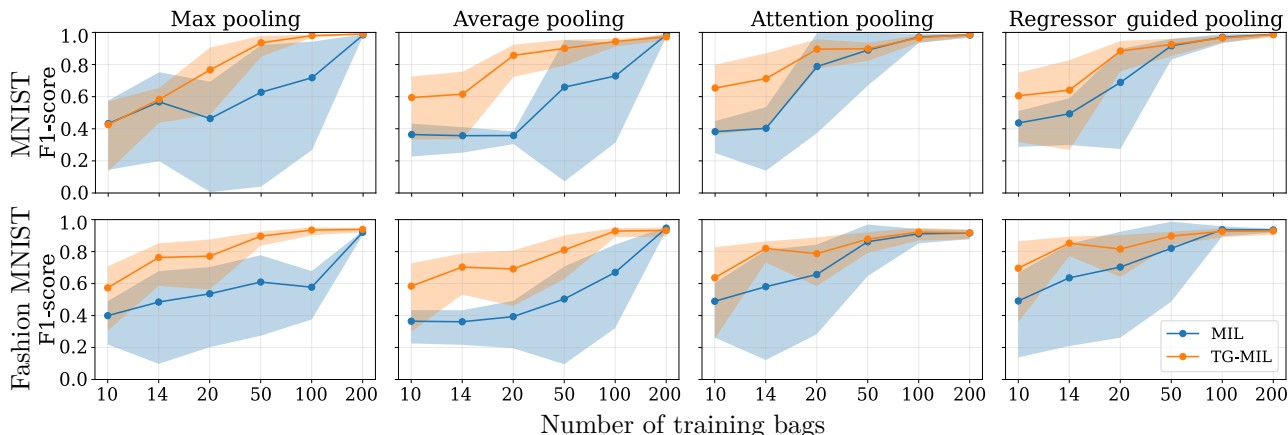

Figure 3: TG-MIL outperforms MIL across different pooling strategies (max, average, attention, and regressor guided) on both MNIST and FashionMNIST datasets. For each number of training bags, the plots show the mean and standard deviation of F1-scores over 5 runs for different bag sizes containing 10, 50, and 100 instances (15 runs in total), highlighting consistent gains especially for smaller training sets.

Table 1: The Wilcoxon rank-sum test shows the significance of topological guidance enhancement over MIL models. Results are reported as p-values for each amount of training data and aggregation technique. P-values less than 0.002 indicate significant enhancement after multiple testing correction(*).

| | Pooling | Number of training bags | | | | | |
| | | 10 | 14 | 20 | 50 | 100 | 200 |
|---|---|---|---|---|---|---|---|
| MNIST | Max | $6.6 \times 10^{-1}$ | $7.7 \times 10^{-1}$ | $3.7 \times 10^{-3}$ | $1.4 \times 10^{-2}$ | $3.8 \times 10^{-2}$ | $4.8 \times 10^{-1}$ |
| | Average | $1.3 \times 10^{-3}*$ | $7.5 \times 10^{-5}*$ | $3.1 \times 10^{-6}*$ | $3.1 \times 10^{-1}$ | $3.7 \times 10^{-1}$ | $1.7 \times 10^{-2}$ |
| | Attention | $7.9 \times 10^{-3}$ | $2.3 \times 10^{-3}$ | $8.0 \times 10^{-1}$ | $3.8 \times 10^{-1}$ | $6.0 \times 10^{-1}$ | $6.5 \times 10^{-1}$ |
| | Regressor | $7.1 \times 10^{-2}$ | $7.8 \times 10^{-2}$ | $6.6 \times 10^{-3}$ | $7.1 \times 10^{-1}$ | $5.1 \times 10^{-1}$ | $3.9 \times 10^{-1}$ |
| Fashion | Max | $1.1 \times 10^{-1}$ | $1.5 \times 10^{-3}*$ | $5.8 \times 10^{-3}$ | $1.3 \times 10^{-2}$ | $1.1 \times 10^{-2}$ | $9.7 \times 10^{-2}$ |
| | Average | $1.2 \times 10^{-1}$ | $2.8 \times 10^{-4}*$ | $9.7 \times 10^{-5}*$ | $2.9 \times 10^{-2}$ | $1.1 \times 10^{-1}$ | $5.6 \times 10^{-1}$ |
| | Attention | $3.8 \times 10^{-1}$ | $5.1 \times 10^{-3}$ | $1.1 \times 10^{-1}$ | $6.5 \times 10^{-1}$ | $2.3 \times 10^{-1}$ | $4.3 \times 10^{-1}$ |
| | Regressor | $9.3 \times 10^{-2}$ | $6.2 \times 10^{-3}$ | $2.8 \times 10^{-1}$ | $3.5 \times 10^{-1}$ | $2.9 \times 10^{-1}$ | $4.1 \times 10^{-1}$ |

ranges from 1 to 1044, providing a comprehensive assessment of our model's adaptability and robustness across different scales of data representation.

**Models.** We use an encoder architecture from the literature to clarify and ensure a fair comparison with existing MIL methods. This architecture includes 2 linear layers with ReLU activations, projecting the input features into a 512-dimensional space for both layers (see Appendix Table 7). For the aggregation function, we opt for regressor-guided pooling (Du et al., 2023). Among all prior methods listed in Table 2, we specifically select RGMIL for reimplementation, as it provides a simpler MIL framework without additional instance-discrimination modules introduced in more recent models such as DistNet (Oner et al., 2023) and APMILwD/GAPMILwD (Zhu et al., 2025). In this way, the only modification to the network architecture of RGMIL in our setup (TG-RGMIL) is to integrate the topological signature calculator into the instances' input and latent spaces.

**Results.** The original RGMIL model used 231 features for FOX, TIGER, and ELEPHANT datasets and 167 features for MUSK1 and MUSK2 datasets, including a last feature representing the repeated label of the bag for each instance. However, in our re-implementation, we follow the standard benchmark settings of 230 and 166 features for the respective datasets to align with previous works and provide a comprehensive comparison (see Table 2). When using this instance feature vector, we observe a decline in the performance of our reimplemented RGMIL baseline, which aligns with the standard benchmark configuration. We run both

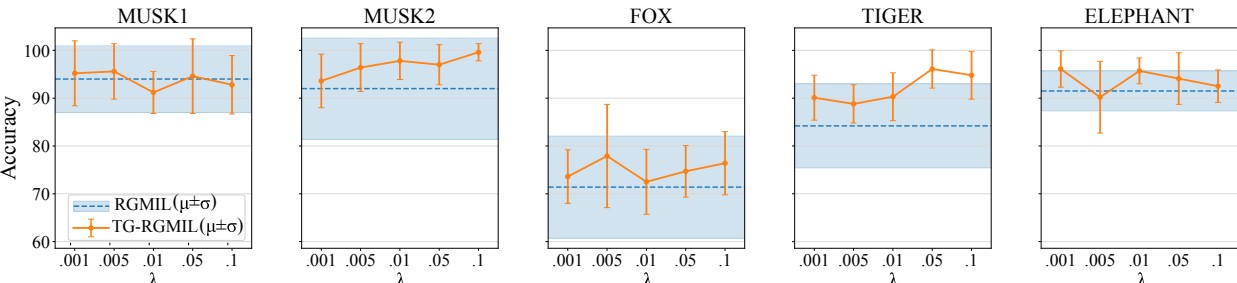

Figure 4: TG-RGMIL accuracy as a function of the topological loss coefficient $\lambda$ on benchmark datasets. Mean $\mu$ and standard deviation ($\sigma$) are computed over five runs.

the RGMIL and TG-RGMIL models 5 times using 10-fold cross-validation and report the average optimal performance. We select the coefficient $\lambda$ heuristically by searching over a range of values compatible with the relative scales of the classification loss $L_{\text{class}}$ and the topological loss $L_{\text{topo}}$, as illustrated in Figure 4 (and Appendix Table 8).

TG-RGMIL consistently outperforms the non-topological approach in all datasets Incorporating higher-dimensional topological features generally improves performance across most cases (MUSK1, FOX, ELEPHANT datasets). For MUSK2 and TIGER, TG-MIL, considering higher-order features, remains competitive with TG-MIL using only 0D topological features, while still outperforming state-of-the-art methods (Table 2). In these two datasets, the underlying instances' topology is likely dominated by connectivity; thus, higher-dimensional topological features do not provide additional class-discriminative cues.

More analysis of these experiments reveals a tendency for the RGMIL model to overfit, which is mitigated by including topological guidance (see Appendix Figure 10). This phenomenon is characterized by the MIL classifier performing best during the initial training epochs.

### 5.3 Anemia Classification

The diagnosis of anemia relies on the presence of a minority of red blood cells in a patient's blood sample that shows morphological deformations associated with the disease. Anemia disorders lead to various aberrant shapes, such as sickle-shaped (SCD), crumpled or perforated (thalassemia), star-shaped (Xero), or spherical (HS) cells. These deformations can manifest with varying degrees of severity and in different proportions, while it is also possible for other cell types unrelated to anemia conditions to coexist.

**Dataset.** Our dataset consists of 521 microscopy images of blood samples obtained from patients who underwent various treatments at different times. Each sample comprises 4 to 12 images, each containing 12 to 45 cells. The data is distributed among five classes: (i) SCD with 13 patients and 170 samples, (ii) Thalassemia with 3 patients and 25 samples, (iii) Xero with 9 patients and 56 samples, (iv) HS with 13 patients and 89, as well as (v) healthy control group consisting of 33 individuals and 181 samples. Similar to previous works, we implement a patient-centric approach by dividing the dataset into three equivalent folds. This division allocates two folds for training and reserves one for testing. This dataset is generated using a pre-trained segmentation method to crop single-cell images (instances) from the whole sample (bag). Cropped images are zero-padded to be the same size, and their encoded extracted features ($4 \times 4 \times 256$) in the segmentation model are used as the input of the MIL instance encoder (see Appendix Table 9). We posit that features extracted by the segmentation model, regardless of cell type, may lack crucial shape information and may even alter the bag's topology. Then we capture the topological reference signature of each bag directly from the image instances, and preserve it in the 500D latent space.

**Models.** For a fair comparison, we apply topological guidance to the previous MIL architecture used for this application, where the instance encoder contains 3 convolutional layers followed by 2 ReLu and Tanh activation functions and a 2 linear layers to obtain a latent representation of instances in a 500D space.

Table 2: Topological guidance improves the classification accuracy (%) of SOTA approaches in MIL benchmarks. Among previous methods, we specifically reimplemented RGMIL for our analysis. Other results (gray) are collected from papers proposed by Ilse et al. (2018) (APMIL and GAPMIL), Yan et al. (2018) (DPMIL), Tu et al. (2019) (GNNMIL), Li et al. (2021) (DSMIL), Huang et al. (2022) (BDRMIL), Oner et al. (2023) (DistNet), Zhu et al. (2025) (APMILwD and GAPMILwD), and Du et al. (2023) (RGMIL). The last three rows correspond to topologically guided RGMIL using increasing levels of topological features: 0D (connected components) only, 0D and 1D (loops), and 0D, 1D, and 2D (voids).

| Method | MUSK1 | MUSK2 | FOX | TIGER | ELEPHANT |
|---|---|---|---|---|---|
| APMIL (2018) | 89.2±4.0 | 85.8±4.8 | 61.5±4.3 | 83.9±2.2 | 86.8±2.2 |
| GAPMIL (2018) | 90.0±5.0 | 86.3±4.2 | 60.3±2.9 | 84.5±1.8 | 85.7±2.7 |
| DPMIL (2018) | 90.7±3.6 | 92.6±4.3 | 65.5±5.2 | 89.7±2.8 | 89.4±3.0 |
| GNNMIL (2019) | 91.7±4.8 | 89.2±1.1 | 67.9±0.7 | 87.6±1.5 | 90.3±1.0 |
| DSMIL (2021) | 93.2±2.3 | 93.0±2.0 | 72.9±1.8 | 86.9±0.8 | 92.5±0.7 |
| BDRMIL (2022) | 92.6±7.9 | 90.5±9.2 | 62.9±11.0 | 86.9±6.6 | 90.8±5.4 |
| DistNet (2023) | 92.3±7.1 | 93.2±6.7 | 68.0±7.5 | 86.4±5.4 | 90.0±7.7 |
| APMILwD (2025) | 96.4±3.3 | 95.4±1.9 | 78.9±4.3 | 91.7±3.6 | 93.4±4.6 |
| GAPMILwD (2025) | 96.7±1.9 | 95.8±2.1 | 78.8±1.6 | 91.9±3.3 | 92.7±3.3 |
| RGMIL (2023) | 94.0±7.0 | 92.0±10.6 | 71.4±10.7 | 84.2±8.8 | 91.5±4.2 |
| TG-RGMIL [0]D | 94.6±7.8 | **97.0±4.2** | 74.7±5.4 | **96.1±4.0** | 94.1±5.4 |
| TG-RGMIL [0,1]D | 96.6±4.8 | 96.7±4.4 | 78.5±5.0 | 95.3±4.1 | 94.2±5.1 |
| TG-RGMIL [0,1,2]D | **98.2±3.4** | 96.2±3.4 | **79.2±5.1** | 95.7±4.2 | **96.3±5.4** |

**Results.** We employ five standard evaluation metrics: Accuracy, F1-Score, Area Under the Receiver Operating Characteristic Curve (AUROC), Precision, and Recall. Recall is reported to assess sensitivity to missed positive cases. All metrics are macro-weighted due to dataset imbalance. Table 3 shows that topological guidance improves the performance of MIL models using *all* aggregation functions and results in higher average performance, often resulting in reduced variance. Topologically guided MIL with *average pooling* surpasses other aggregation schemes. This aligns with our findings from experiments on synthetic datasets, where we observe that topological guidance particularly narrows the gap in performance between MILs employing different aggregation functions. The inherent ambiguity in the anemia dataset for MIL suggests that enhancing instance projection in latent space via average pooling is more effective than attention pooling, as it better captures the ratio of positive instances. Without topological guidance, scarce training data impede the instance encoder from generating meaningful, generalizable latent representations. However, integrating topological inductive bias into the latent space mitigates these challenges, significantly improving MIL performance. $\lambda = 0.005$ provides the best overall performance across pooling strategies. Notably, the performance is relatively stable across 2 orders of magnitude of $\lambda$ (see Figure 5 and Appendix Table 10).

**Instance-level analysis.** Figure 6 shows anomaly scores with and without topological guidance. Without topological guidance, the anomaly detector assigns inconsistent scores to visually similar instances, indicating instability in instance representation learning rather than instance-level diagnostic reliability. This inconsistency is mitigated by topological guidance, highlighting a challenge in uniformly evaluating similar data points and demonstrating its effectiveness in improving representation consistency and interpretability. Further illustrating this point, we visualize the distance matrix of instances within a bag in the input space and compare them with their corresponding matrices in the latent space, both with and without topological guidance. Figure 7 shows that MIL with topological guidance better preserves the relative distances between bag instances in the latent space projection. In contrast, MIL without topological guidance selects a limited number of instances and pushes them far away from the majority in the latent space. This behavior suggests

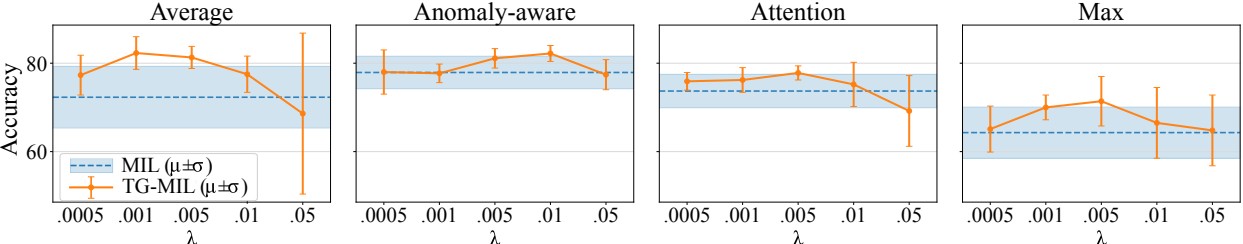

Figure 5: Effect of the topological loss weight $\lambda$ on TG-MIL performance for the anemia classification task. Accuracy is reported for four pooling strategies across five $\lambda$ values. $\mu$ and $\sigma$ indicate the average and standard deviation of accuracy across 3 runs.

Table 3: Topological guidance improves classification performance (%) for all pooling strategies in Anemia classification. We apply it to different MIL methods with various pooling techniques containing Average, Anomaly-aware (Kazeminia et al., 2022), Attention (Sadafi et al., 2020), and Max pooling. Numbers depict average classification performance and standard deviation from 3 cross-validation runs. Best performance is indicated by bold text. We compare classification performance without (✗) and with (✓) topological guidance, with the superior result underlined.

| Pooling | Average | | Anomaly-aware | | Attention | | Max | |
|---|---|---|---|---|---|---|---|---|
| $L_{\text{topo}}$ | ✗ | ✓ | ✗ | ✓ | ✗ | ✓ | ✗ | ✓ |
| Accuracy | 72.3±7.0 | **81.3±2.5** | 77.9±3.7 | 81.1±2.2 | 73.7±3.8 | 77.8±1.6 | 64.3±5.8 | 71.4±5.6 |
| F1-Score | 70.5±7.4 | **80.3±3.1** | 76.7±4.0 | 79.2±2.6 | 72.4±3.8 | 74.7±1.6 | 63.0±5.0 | 68.8±5.4 |
| AUROC | 89.9±2.7 | **93.7±4.4** | 89.1±4.3 | 93.1±4.0 | 91.6±3.0 | 91.9±2.5 | 84.8±2.8 | 89.7±3.1 |
| Recall | 59.7±7.8 | **65.1±5.0** | 63.2±3.2 | 65.9±0.7 | 59.3±6.4 | 60.4±2.3 | 52.8±8.6 | 53.8±4.7 |
| Precision | 61.8±7.1 | **79.1±12.0** | 67.4±4.5 | 69.6±0.6 | 64.9±4.9 | 73.2±7.9 | 52.9±9.6 | 63.4±7.5 |

that, without topological guidance, only a few instances dominate the bag representation, explaining the observed instability in instance-level scores.

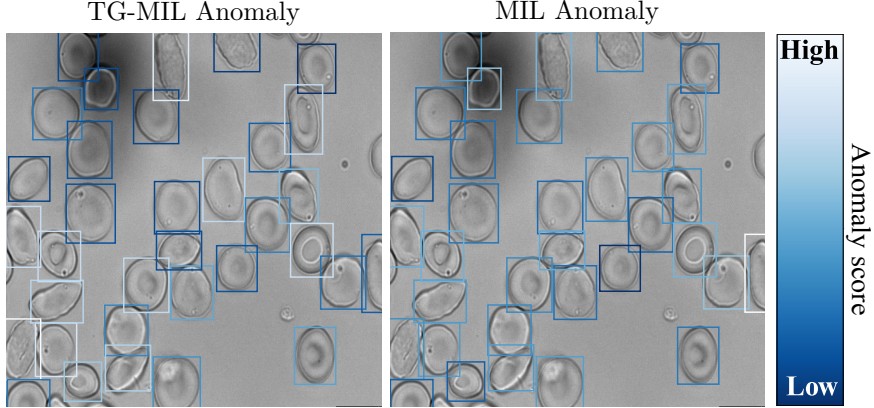

Figure 6: Topological guidance enhances the model's ability to identify disease-relevant (sickle) cells more effectively. TG-MIL results in more uniform anomaly scores for deformed cells, in contrast to the varied scores resulting from MIL Anomaly.

In addressing potential inquiries about our choice of topological guidance over a distance-preservation-based loss, it is worth noting the distinct advantages of our approach. Topological loss is particularly robust to noise

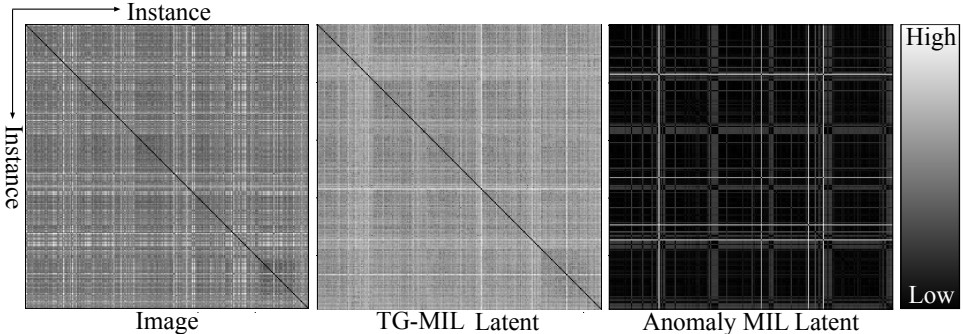

Figure 7: Heatmaps of distances between instance images (left), TG-MIL latent vectors (middle), and anomaly-aware MIL latent vectors (right) for the same bag. Topological guidance preserves the topological characteristics of the image space, making the latent space more closely resemble the original high-dimensional space.

and effective in high-dimensional spaces, and it exhibits scale invariance, enabling preservation of relative distance patterns among instances within a bag. These properties allow the model to maintain meaningful instance relationships in the latent space, thereby supporting more stable bag-level predictions and qualitative interpretability, rather than instance-level diagnostic conclusions.

### 5.4 Unit Test

We evaluate TG-MIL using the unit test introduced by Raff & Holt (2023), which serves as a functional check to verify that a MIL model does not exploit invalid shortcuts when predicting bag-level labels.

**Concept.** The unit test follows the classical single-concept MIL formulation of Dietterich et al. (1997), in which bag labels are determined by the presence of a single positive concept. The concept class is unitary, consisting of a positive concept $c_1$ and a null concept $\emptyset$. A bag is labeled positive if and only if at least one instance corresponding to $c_1$ is present. Each bag contains instances sampled from simple Gaussian distributions. Most instances are negative instances, drawn from a background distribution $\mathcal{N}(0, I_d)$, which corresponds to the null concept and carries no label information. Positive bags additionally include a small number of informative instances drawn from one of two non-co-occurring distributions, $\mathcal{N}(0, 3I_d)$ or $\mathcal{N}(1, I_d)$, both mapped to the same positive concept $c_1$. As no single positive pattern is shared across all positive bags, correct prediction requires aggregating evidence across instances rather than detecting a fixed prototype. The unit test further introduces a bait distribution $\mathcal{N}(-10, 0.1I_d)$, which is trivially separable from all other instances. During training, a poison instance from the bait distribution appears exclusively in negative bags, creating an invalid but easier-to-learn decision rule based on the absence of the bait instance. This rule violates the MIL assumption that bag labels depend on the presence of positive concepts. Models that learn this MIL-violating shortcut achieve high training performance but generalize incorrectly at test time, where the bait–label correlation is reversed. Following the authors' criterion, a model is considered to fail the unit test if it achieves training AUC > 0.5 but test AUC < 0.5, indicating learning an anti-correlated, invalid decision rule rather than the intended existential MIL rule.

**Results.** Table 4 reports the unit-test results of TG-MIL under different pooling operators.

TG-MIL with average pooling achieves strong generalization, with a test balanced accuracy of 0.90 and an AUC of 0.90, indicating reliable recovery of the intended existential MIL decision rule. Both attention-based and regressor-guided pooling pass the unit test according to the authors' criterion, achieving test AUCs of 0.91 and 0.90, respectively. However, their test balanced accuracies (0.74 and 0.47) are substantially lower than average pooling, indicating less stable decision behavior under the test distribution and greater sensitivity to spurious correlations. Max pooling fails the unit test, yielding a test-balanced AUC of 0.50, corresponding

Table 4: Unit-test results of TG-MIL with different pooling operators.

| Pooling Method | Train AUC | Test Acc. | Test AUC |
|---|---|---|---|
| Average Pooling | 0.99 | 0.90 | 0.90 |
| Max Pooling | 0.98 | 0.50 | 0.50 |
| Attention Pooling | 1.00 | 0.74 | 0.91 |
| Regressor-guided Pooling | 0.99 | 0.47 | 0.90 |

to random-guessing behavior. While topological inductive bias improves max pooling compared to a MIL baseline (increasing test AUC from 0.00 to 0.50), this remains insufficient to pass the test.

### 5.5 Practical Computational Cost

We empirically evaluate the computational overhead introduced by the topological signature calculation using two complementary timing metrics: (i) the average training time per iteration, which reflects the end-to-end cost of one optimization step, including forward propagation, loss computation, and backpropagation, and (ii) the wall-clock forward time, defined as the real elapsed time during a single forward pass. The baseline MIL model requires 0.021 seconds per iteration, whereas TG-MIL requires 0.077 seconds per iteration, a $3.7\times$ increase in end-to-end training time per iteration. For a bag of size $n = 200$, the wall-clock forward time increases from 0.15 seconds for MIL to 0.35 seconds for TG-MIL, a 2.33 fold increase and a $\Delta t(200) = 0.20$ seconds. For a bag of size $n = 100$, the corresponding increase is from 0.12 seconds to 0.18 seconds (1.53 fold change, $\Delta t(100) = 0.059$ seconds). When the bag size doubles from 100 to 200, the additional forward-time overhead increases by a factor of approximately 3.5, which is close to the factor of 4 predicted by the quadratic complexity analysis. Since TG-MIL introduces no additional learnable parameters, these observations support the quadratic dominant term identified in the theoretical analysis.

**CO2 Emission Related to Experiments.**  All experiments were conducted for 228 hours on institute's infrastructure with $0.432$ kgCO$_2$eq/kWh carbon efficiency and A100 PCIe 40/80GB hardware (TDP 250W), resulting in $24.62$ kgCO$_2$eq total emissions with no direct offset. Calculations utilized the MachineLearning Impact calculator (Lacoste et al., 2019).

## 6 Conclusion

We introduce TG-MIL, a novel approach that leverages the topological properties of input bags as an inductive bias for their latent representation in MIL. This is achieved by employing a topological loss term within the MIL objective function. Our method ensures that the intrinsic geometric-topological characteristics of bags in the input space are consistently preserved in their latent projections. Testing our approach across different datasets, we show that preserving data topology in latent MIL models leads to substantial improvements in predictive and generalization performance, especially when dealing with scarce training data. While the datasets used in our experiments involve non-trivial MIL problems, the individual instances (images) exhibit relatively simple visual structures, e.g., centered, grayscale, background-free. In such settings, pixel-level differences provide an effective reference for defining topology in the latent space. However, in more visually complex domains, where instances are more complex, e.g., with greater diversity of textures, colors, or higher background noise, pixel-based topology may become less reliable, as it can transfer irrelevant visual noise into the latent representation. The robustness of TG-MIL on benchmark datasets, where the original images were unavailable and only pre-extracted image features were used as inputs, further suggests that defining topology over higher-level feature representations is a viable alternative. This observation does not imply a direct comparison with image-based inputs but rather highlights the potential of feature-level topology as a more general inductive bias for visually complex image instances.

As for future research directions, we plan to explore alternative methods for describing image geometry and topology, with a particular focus on cubical complexes that can directly operate on images. Additionally, we aim to investigate the geometrical and topological properties of bag spaces, leveraging recent advances in

metric geometry, including the Gromov–Hausdorff distance, which has previously been utilized to characterize shapes in related studies (Chazal et al., 2009).

**Broader Clinical Impact Statement**

Notably, in the current clinical setting, AI tools based on our method should be used strictly as decision-support systems for clinicians, rather than as standalone diagnostic oracles, due to the weakly supervised nature of MIL, potential misclassifications, data distribution shifts, and clinical liability considerations.

**Acknowledgments**

We gratefully acknowledge Anna Bogdanova, Asya Makhro, and Ario Sadafi for providing the biomedical dataset used in this research. The anemia dataset is not publicly available. Access may be granted upon reasonable request to the data owners (Sadafi et al., 2020).

The Helmholtz Association supports the present contribution under the joint research school "Munich School for Data Science - MUDS". C.M. has received funding from the European Research Council (ERC) under the European Union's Horizon 2020 research and innovation program (Grant Agreement No. 866411). B.R. was partially supported by the Bavarian state government with funds from the *Hightech Agenda Bayern*.

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

# A   Appendix

## A.1   Toy experiment

Table 5 specifies more details of the MIL architecture we developed to classify toy dataset.

| Component | Configuration |
|---|---|
| Input channels | 100 |
| Instance encoder | Linear($100 \rightarrow 64$), ReLU, Linear($64 \rightarrow 2$), ReLU |
| Latent dimension | 2 |
| Pooling | Regressor guided pooling |
| Classifier head | Linear ($100 \rightarrow 2$) |
| Loss | BCEWithLogitsLoss and $L_{topo}$ |
| Optimizer | Adam (lr: 0.0005) |
| $\lambda$ | 0.005 |

Table 5: Architecture of the MIL model for the toy experiment.

## A.2   Synthetic data experiments

For synthetic data we employed same architecture for both MIL-MNIST and MIL-FashionMNIST datasets. We explored different aggregation functions containing max pooling,

$$\zeta_{b_m} = \max_{i \leq n} z_i, \tag{11}$$

average pooling,

$$\zeta_{b_m} = \frac{\sum_{i=1}^{n} z_i}{n}, \tag{12}$$

and attention pooling with parameters $W$ and $V$

$$\zeta_{b_m} = \sum_{i=1}^{n} a_i z_i, \tag{13}$$

where

$$a_i = \frac{\exp(W^T \tanh(V z_i^T))}{\sum_{i=1}^{n} \exp(W^T \tanh(V z_i^T))}. \tag{14}$$

The RGMIL (Du et al., 2023) model uses regressor guided pooling technique. In this model, the regressor (with parameters $W$ and $B$) calculates the binary probability value of instance latent representation $z_i$ as

$$(p_i^+, p_i^-) := W^T z_i + B, \tag{15}$$

where $p_i^+$ and $p_i^-$ show the probability of instance $z_i$ to belong to the positive or negative class. Then it gets the difference between these two achieved probabilities,

$$p_i = p_i^+ - p_i^-, \tag{16}$$

normalizes the result

$$\omega_i = \frac{p_i - \mathbb{E}[p_i]}{\sqrt{Var(p_i)}}, \tag{17}$$

and applies softmax on it

$$\alpha_i = \frac{\exp(\omega_i)}{\sum_{j=1}^{n} \exp(\omega_j)}. \tag{18}$$

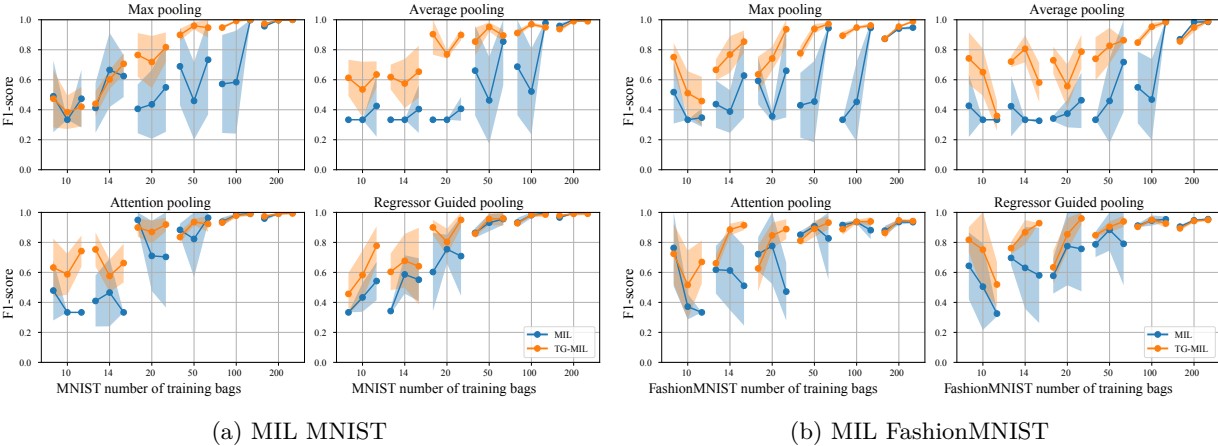

(a) MIL MNIST          (b) MIL FashionMNIST

Figure 8: TG-MIL outperforms MIL models irrespective of the aggregation function when subjected to a limited amount of training bags. For each number of training bags, the average and standard deviation for the F1-score of the model's performance in 5 runs for each bag size of 10, 50, and 100 (in total 15 runs) is shown.

where, $\alpha_i$ specifies the pooling weight of $z_i$. Then the latent of bag is

$$\zeta_{b_m} = \sum_{i=1}^{n} \exp(\alpha_i z_i). \tag{19}$$

Table 6 shows the settings of TG-MIL architecture architecture used for synthetic dataset classification.

| Parameter | Max Pooling | Average Pooling | Attention Pooling | RGP Pooling |
|---|---|---|---|---|
| Pooling | Max | Average | Attention | Regressor guided |
| In dimension | | | $28 \times 28$ | |
| Input channel | | | 1 | |
| Instance encoder | | Linear(1→20), ReLU, Linear(20→50), ReLU, Linear(50→500), ReLU | | |
| Latent dimension | | | 500 | |
| Attention latent dimension | | | 128 | |
| Linear layer | | | $500 \times 2$ | |
| Loss | | BCEWithLogitsLoss and $L_{topo}$ | | |
| Optimizer | Adam (LR: 0.005) | | Adam (LR: 0.0005) | |
| Batch size | | | 1 | |
| Max epochs | | | 100 | |

Table 6: General architecture and configurations of TG-MIL Model for synthetic datasets over different pooling strategies. The value $\lambda$ is not reported here as it differs between datasets, training budgets, and bag sizes. Please find its relevant value in the source code.

Figure 8 distinctly illustrates how topological guidance addresses data scarcity issue in MIL considering different scenarios, containing different amount of training data and bag sizes.

Figure 9 display the learning curves of training RGMIL alongside TR-RGMIL on synthetic dataset. For RGMIL, the loss corresponds to the bag-level cross-entropy loss, while for TG-RGMIL, it is the total loss, defined as the sum of the cross-entropy (CE) and topological losses. Since the topological loss operates at a larger scale, the total loss magnitude for TG-RGMIL is higher, particularly during the early stages of training. Despite the different loss scales, TG-RGMIL consistently converges to a lower final total loss, indicating that the cross-entropy component also converges to a lower value compared to RGMIL, making the two approaches comparable at convergence. The introduction of topological giudance addresses overfitting in RGMIL and results in a significant improvement in its classification performance.

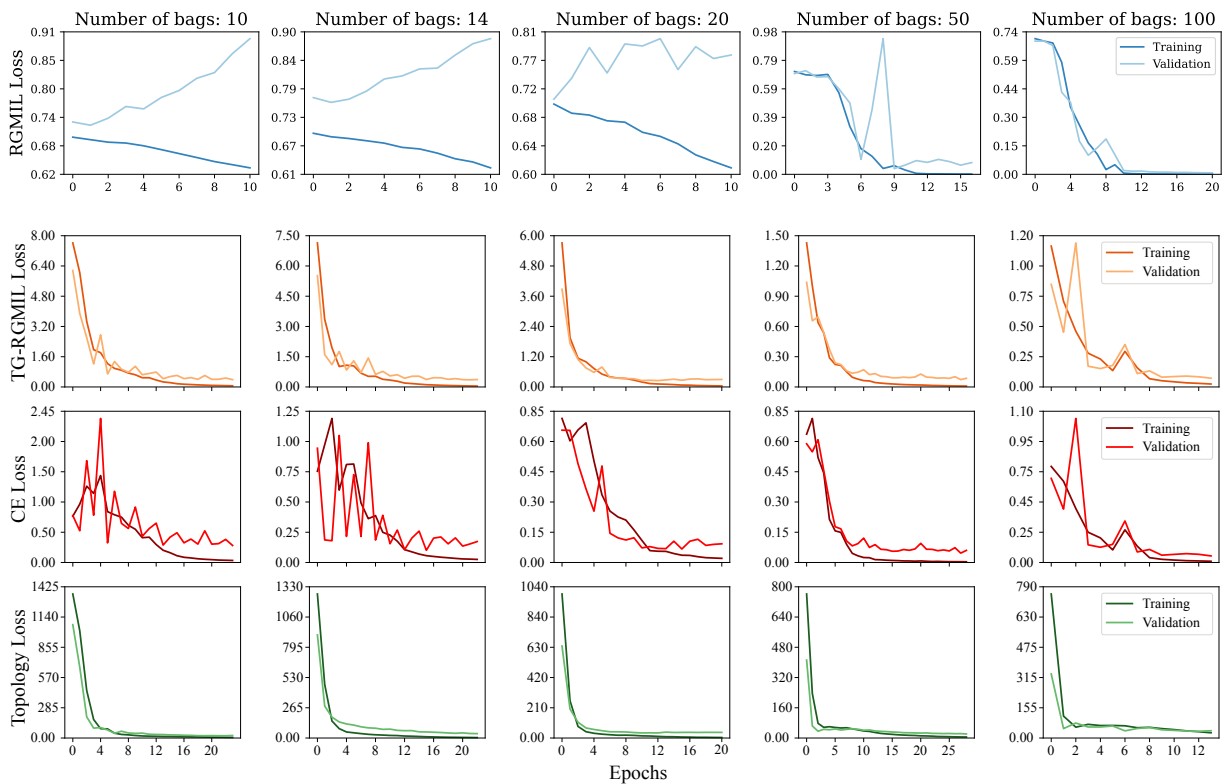

Figure 9: Topological guidance enhances the RGMIL model generalizability for scarce training data. Each column displays the learning curves of models trained with 10 bags, each containing approximately 10 instances on average.

## A.3 Benchmark experiments

Table 7 shows the settings of TG-MIL architecture architecture used for benchmark experiments.

| Components | Elephant, Fox, Tiger | Musk1, Musk |
|---|---|---|
| Input channels | 230 | 166 |
| Instance encoder | Linear(231, 512), ReLU, Linear(512, 512), ReLU | |
| Latent Dimension | 512 | |
| Pooling | Regressor guided pooling | |
| Classifier head | Linear $(512 \rightarrow 2)$ | |
| Loss | BCEWithLogitsLoss and TopoRegLoss | |
| Optimizer | Adam (lr: 0.00005, Betas: $[0.9, 0.999]$) | |
| Max Epochs | 40 | |
| $\lambda$ | 0.05 | |

Table 7: Architecture of the MIL Model for Benchmarks

Table 8 shows the sensitivity of TG-RGMIL to $\lambda$ coefficient for benchmark datasets.

Figure 10 shows how topological guidance helps with overfitting of the RGMIL model tends on benchmark datasets.

Table 8: TG-RGMIL accuracy (%) across different values of $\lambda$ (mean $\pm$ std) for Benchmark datasets.

| Dataset | $\lambda = 0.001$ | $\lambda = 0.005$ | $\lambda = 0.01$ | $\lambda = 0.05$ | $\lambda = 0.1$ |
|---|---|---|---|---|---|
| MUSK1 | $95.2 \pm 6.8$ | $95.6 \pm 5.8$ | $91.2 \pm 4.4$ | $94.6 \pm 7.8$ | $92.8 \pm 6.1$ |
| MUSK2 | $93.6 \pm 5.6$ | $96.4 \pm 5.0$ | $97.8 \pm 3.9$ | $97.0 \pm 4.2$ | $99.6 \pm 1.8$ |
| FOX | $73.6 \pm 5.6$ | $77.9 \pm 10.8$ | $72.5 \pm 6.8$ | $74.7 \pm 5.4$ | $76.4 \pm 6.6$ |
| TIGER | $90.1 \pm 4.7$ | $88.8 \pm 4.0$ | $90.3 \pm 5.0$ | $96.1 \pm 4.0$ | $94.8 \pm 5.0$ |
| ELEPHANT | $96.1 \pm 3.8$ | $90.2 \pm 7.5$ | $95.7 \pm 2.7$ | $94.1 \pm 5.4$ | $92.5 \pm 3.4$ |

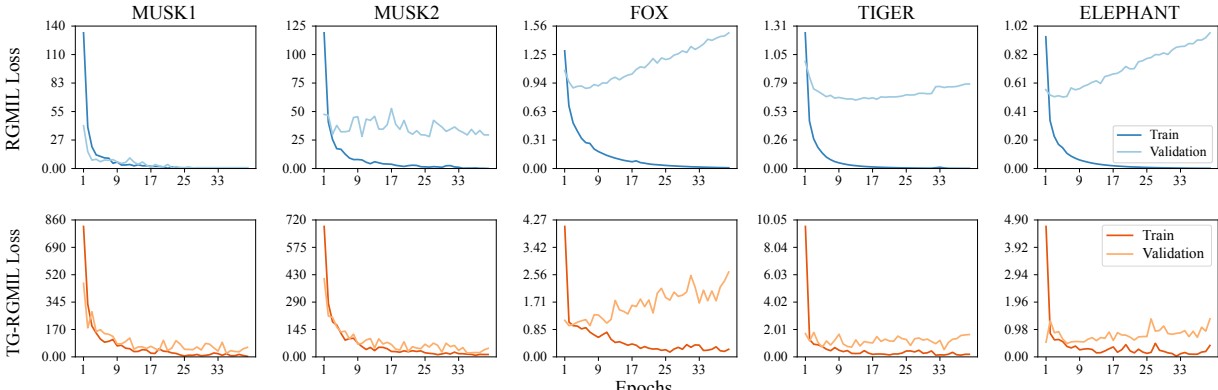

Figure 10: Topological guidance enhances RGMIL model generalizability for Benchmarks. Each column shows learning curves achieved by a MIL benchmark dataset.

### A.4 Anemia experiments

For anemia classification we followed the architecture of the state of the art in this application (Kazeminia et al., 2022) (Table 9). This method introduced anomaly score to be considered in addition to attention values to estimate the importance of each instance. To this end the distribution of negative instances is estimated from negative bags by fitting a gaussian mixture model on their latent representation. The anomaly score of each instance latent $z_i$ is calculated as

$$d_i = \sqrt{(z_i - \mu)^T \Sigma^{-1} (z_i - \mu)}, \tag{20}$$

where $\mu$ and $\Sigma$ are mean and covariance of the fitted GMM on negative distribution. Then the pooling weight of the instance is calculated as a linear combination of attention score $a_i$ and anomaly score $d_i$. With this the bag latent is

$$\zeta_{b_m} = \sum_{i=1}^{n} (W_{D_i} d_i + W_{A_i} a_i) z_n. \tag{21}$$

The other consideration of this approach is the formulation of $Loss_{class}$ with a dual classifier head (Sadafi et al., 2020) that comprises a bag classifier head and an instance classifier head. The bag classifier head is trained using a cross-entropy loss function $L_{\text{bag}}$, calculated as the difference between the predicted bag label and the corresponding ground truth label for the bag. The instance classifier head is trained using a cross-entropy loss function $L_{\text{Instance}}$ that utilizes the noisy labels of instances as the repeated labels of the bag for all instances. The final MIL classification loss is calculated as

$$L_{class} = (1 - \gamma) L_{\text{bag}} + \gamma L_{Instance}, \tag{22}$$

where $\gamma$ is a coefficient that decreases as with epoch number increasing.

| Parameter | Max Pooling | Average Pooling | Aux Attention | Anomaly Detection |
|---|---|---|---|---|
| Image dimentions | | $64 \times 64$ | | |
| Image channels | | 1 | | |
| Features channels | | 256 | | |
| Instance encoder | | Conv2D(256→301), ReLU, Conv2D(301→500), ReLU, Conv2D(500→650), Tanh, Linear(650→500)) | | |
| Latent dimension | | 500 | | |
| Instance classifier head | | - | Linear(500 → 500), Linear(500 → 5) | |
| Pooling | Max | Average | Attention | Anomaly |
| Attention layer | | - | Linear(500→128), Tanh, Linear(128→1) | |
| Bag classifier head | | Linear(500 → 2) | | |
| Loss | bag CrossEntropyLoss and TopoRegLoss | | CrossEntropyLoss (bag and instance) and TopoRegLoss | |
| Optimization | | Adam (lr=0.0005 | | |
| Learning rate | | 0.0005 | | |
| Max epochs | | 300 | | |
| Early stopping | | patience: 50 | | |
| Image input channels | | 1 | | |
| $\lambda$ | | 0.005 | | |

Table 9: Configuration of MIL model for anemia classification with different pooling strategies

Table 10: TG-MIL accuracy (%) across different values of $\lambda$ (mean $\pm$ std) over different aggregations for anemia dataset.

| Pooling Method | $\lambda = 0.0005$ | $\lambda = 0.001$ | $\lambda = 0.005$ | $\lambda = 0.01$ | $\lambda = 0.05$ |
|---|---|---|---|---|---|
| Average | $77.3 \pm 4.5$ | $82.3 \pm 3.7$ | $81.3 \pm 2.5$ | $77.5 \pm 4.1$ | $68.6 \pm 18.2$ |
| Max | $65.1 \pm 5.2$ | $70.0 \pm 2.8$ | $71.4 \pm 5.6$ | $66.5 \pm 8.0$ | $64.8 \pm 8.0$ |
| Attention | $75.9 \pm 2.0$ | $76.2 \pm 2.8$ | $77.8 \pm 1.6$ | $75.2 \pm 5.0$ | $69.2 \pm 8.0$ |
| Anomaly-aware | $78.0 \pm 5.0$ | $77.7 \pm 2.1$ | $81.1 \pm 2.2$ | $82.2 \pm 1.8$ | $77.4 \pm 3.4$ |

