# OpenReview forum: "Topological Inductive Bias fosters Multiple Instance Learning in Data-Scarce Scenarios"
_TMLR — Accepted by TMLR_

### Review · Reviewer_B9LH · 2025-11-28

**Summary Of Contributions:**

The paper proposes **Topology Guided Multiple Instance Learning (TG-MIL)**, a novel framework that incorporates a **topology-preserving inductive bias** into the MIL latent space. This constraint acts as a powerful regularizer, ensuring the intrinsic structural distribution of instances within a bag is maintained when mapped to the latent space. The primary contribution is providing an effective solution for MIL's sharp performance drop in **data-scarce scenarios** (e.g., rare disease classification), showing consistent and substantial empirical improvements.

#### Key Strengths:

1.  **Novelty:** Innovative integration of **Topological Data Analysis (TDA)** as a structural inductive bias for generalization.
2.  **Practical Relevance:** Directly addresses the critical challenge of data scarcity in high-stakes applications.
3.  **Strong Performance:** Reports significant performance gains (up to 15.3% on synthetic and 5.5% on rare disease data) under limited training data conditions.

#### Key Weaknesses:

1.  **Methodological Opacity:** Lacks clear mathematical formulation and definition of the specific TDA metrics used for the topological loss ($L_{\text{topo}}$).
2.  **Fundamental Property Concern:** **Critically, the paper omits theoretical proof and empirical unit testing** to verify the method maintains the essential MIL property of **instance permutation invariance**.
3.  **Unquantified Cost:** Computational overhead from TDA calculation is not adequately measured or discussed.

**Additional Comments:**

None

**Audience:**

No

**Audience Explanation:**

The findings are of high interest to several segments of the TMLR audience due to the novel methodology and significant practical impact:

1.  **MIL Researchers:** The paper introduces a novel, principled regularization technique, offering a new structural **inductive bias** for the MIL latent space.
2.  **Topological Machine Learning (TML) Community:** It provides a strong, practical example of how **Topological Data Analysis (TDA)** techniques can be effectively integrated as a loss term to solve complex machine learning problems.
3.  **Applied AI/Medical Practitioners:** The demonstrated ability to achieve superior results in high-stakes, **data-limited fields** (like rare disease classification) is highly valuable to application-focused readers.

**Broader Impact Concerns:**

1.  **Safety and Error Modes (False Negatives):** In rare disease diagnosis, a **False Negative** (missing a diagnosis) can have catastrophic consequences for the patient. The authors must discuss the model's **robustness** and potential **failure modes** regarding high-risk outcomes. Specifically, they should comment on how their method controls or impacts the False Negative rate compared to baselines, even when data is scarce.
2.  **Dataset Bias and Generalizability:** The method is tested on specific rare anemia datasets. Any inherent bias in the data source (e.g., patient demographics, imaging equipment/protocols from a specific clinic, or geographical location) could mean the learned topological bias is not truly generalizable. The authors should acknowledge these potential **domain shifts** and discuss how the model's performance might degrade when deployed to external hospitals or patient populations.
3.  **Misinterpretation of Instance Predictions:** The method seeks to improve instance representations. This encourages the use of the model for **interpretable localization** (identifying which instance is "pathological"). If the instance-level predictions are used for diagnosis, the authors must caution against over-reliance, as MIL is inherently a weak supervision framework. The Broader Impact Statement should advocate for the model to be used as a **decision support tool** for clinicians, not as a standalone diagnostic oracle.

**Claims And Evidence:**

Yes

**Claims Explanation:**

The claims are convincingly supported by the evidence, particularly regarding efficacy in **data-scarce regimes**.

1.  **Targeted Validation:** Results (Figures 6, 7) consistently show that TG-MIL significantly outperforms baselines when trained with a **limited number of bags**. This empirically validates the method's core goal of addressing data scarcity.
2.  **Generalizability Confirmed:** The topological constraint acts as an effective regularizer, helping models like RGMIL **mitigate overfitting** on small benchmark datasets (Figure 8), thereby supporting the claim of enhanced generalizability.
3.  **Practical Proof:** Success on the rare anemia dataset provides strong real-world evidence of the method's utility in high-stakes, data-limited applications.

**Requested Changes:**

It is a good paper. However, these changes are fundamental to the algorithmic correctness and rigorous presentation required for an MIL submission. **The novelty of integrating a topological constraint (TDA) introduces a necessary burden of proof to ensure foundational MIL properties are preserved.**

1.  **Rigorous Verification of Instance Permutation Invariance:**
    * **The authors must explicitly address the uncertainty regarding whether the introduction of $L_{\text{topo}}$ alters the inherent permutation invariance of the standard MIL aggregation mechanism.**
    * **Theoretical Proof:** Provide a formal mathematical argument in the main text demonstrating that the total loss function ($L_{\text{total}} = L_{\text{MIL}} + \lambda L_{\text{topo}}$) is invariant to any arbitrary permutation of instances within a bag.
    * **Empirical Unit Test:** Include an explicit algorithmic unit test, as recommended by [2]. This test must empirically show that randomly permuting instances within a fixed test bag does not change the model's predicted bag label and confidence score. This is non-negotiable for a valid MIL algorithm.

2.  **Detailed Methodological Formulation:**
    * Provide the **exact mathematical formulation** of the topology-preserving constraint/loss function $L_{\text{topo}}$. This must clearly specify:
        * Which **Topological Data Analysis (TDA) features** are computed (e.g., Persistence Diagrams, Betti numbers).
        * The **distance metric** used to compare the topology between the original and latent spaces (e.g., Wasserstein or Bottleneck distance).

3.  **Contextualization with Instance Learnability:**
    * Include a discussion section that **cites and integrates the findings of [1]**. Authors should try to explain how the topological inductive bias specifically enhances the **learnability and robustness of instance-level representations**, thereby indirectly aiding instance-level inference derived from the bag labels.

---

**References:**

[1] Are Multiple Instance Learning Algorithms Learnable for Instances? NeurIPS, 2022.

[2] Raff E., & Holt J. Reproducibility in Multiple Instance Learning: A Case for Algorithmic Unit Tests. NeurIPS, 2023.

---

> ### Author Response · Authors · 2025-12-25
>
> **Requested Change 1**:
>
> **[Theoretical Proof]**: We thank the reviewer for highlighting this important point. We add the following clarification in the Method section:
>  >The topological loss is defined in terms of pairwise distances between instances within a bag. Any permutation of instances changes only their ordering and leaves the set of pairwise distances unchanged. Under the assumption that pairwise distances are unique, their relative ordering is also preserved. Consequently, the induced topological loss is invariant to arbitrary permutations of instances.
>
> **[Empirical Unit Test]**: We appreciate the reviewer's suggestion. Accordingly, we applied the suggested unit test to TG-MIL across four aggregation functions. The results showed that TG-MIL passes this test successfully with a test AUC of > 0.05 when used with average pooling, attention-based pooling, and regressor-guided pooling within the MIL framework. We add these results to our manuscript with the following text and table.
> > ### **Unit Test**
> >
> >We evaluate TG-MIL using the unit test introduced in Raff & Holt (2023), which is designed as a functional check to verify that a MIL model does not exploit invalid shortcuts when predicting bag-level labels.
> >
> >**Concept.** The unit test follows the classical single-concept MIL formulation of Raff & Holt (2023),  where bag labels are determined by the existence of a single positive concept. The concept class is unitary, consisting of a positive concept $c\_1$ and a null concept $\emptyset$. A bag is labeled positive if and only if at least one instance corresponding to $c\_1$ is present. Each bag contains instances sampled from simple Gaussian distributions. Most instances are negative instances, drawn from a background distribution $\mathcal{N}(0, I\_d)$, which corresponds to the null concept and carries no label information. Positive bags additionally include a small number of informative instances drawn from one of two non-co-occurring distributions, $\mathcal{N}(0, 3I\_d)$ or $\mathcal{N}(1, I\_d)$, both mapped to the same positive concept $c\_1$. As no single positive pattern is shared across all positive bags, correct prediction requires aggregating evidence across instances rather than detecting a fixed prototype. The unit test further introduces a bait distribution $\mathcal{N}(-10, 0.1 I\_d)$, which is trivially separable from all other instances. During training, a poison instance from the bait distribution appears exclusively in negative bags, creating an invalid but easier-to-learn decision rule based on the absence of the bait instance. This rule violates the MIL assumption that bag labels depend on the presence of positive concepts. Models that learn this MIL-violating shortcut achieve high training performance but generalize incorrectly at test time, where the bait–label correlation is reversed. Following the authors’ criterion, a model is considered to fail the unit test if it achieves training AUC $> 0.5$ but test AUC $< 0.5$, indicating learning an anti-correlated, invalid decision rule rather than the intended existential MIL rule.
> >
> >**Results.** Table 4 reports the unit-test results of TG-MIL under different pooling operators. TG-MIL with average pooling achieves strong generalization, with a test balanced accuracy of 0.90 and an AUC of 0.90, indicating reliable recovery of the intended existential MIL decision rule. Both attention-based and regressor-guided pooling pass the unit test according to the authors’ criterion, achieving test AUCs of 0.91 and 0.90, respectively. However, their test balanced accuracies (0.74 and 0.47) are substantially lower than average pooling, indicating less stable decision behavior under the test distribution and greater sensitivity to spurious correlations.
> Max pooling fails the unit test, yielding a test-balanced AUC of 0.50, corresponding to random-guessing behavior. While topological inductive bias improves max pooling compared to a MIL baseline (increasing test AUC from 0.00 to 0.50), this remains insufficient to pass the test.
> >
> >| TG-MIL Pooling Method        | Train AUC | Test Acc. | Test AUC |
> >|-----------------------------|-----------|-----------|----------|
> >| Average Pooling              | 0.99      | 0.90      | 0.90     |
> >| Max Pooling                  | 0.98      | 0.50      | 0.50     |
> >| Attention Pooling            | 1.00      | 0.74      | 0.91     |
> >| Regressor-guided Pooling     | 0.99      | 0.47      | 0.90     |

---

> ### Author Response · Authors · 2025-12-25
>
> **Requested Change  2**: We thank the reviewer for raising this important point, which we addressed in the following ways:
> With respect to TDA features: In the Method section, paragraph Complexity and Parameters, we mentioned that we use persistent diagrams on 0-dimensional topological features (connected components). Notably, we do not employ Betti curve or any other TDA feature representation techniques to represent the persistent diagram. For clarification, we rewrote the following paragraph of the Method section, which now reads:
>
> >We calculate the simplicial complex of Vietoris-Rips (${\mathfrak{R}}\_{\epsilon}(X)$) at each distance $\epsilon$, which yields all subsets of $X$ such that the pairwise distance between points is less than or equal to $\epsilon$.
> >
> >$${\mathfrak{R}}\_{\epsilon}(X\_{b\_m}):= \{x\_i| \exists x\_j \in X , x\_i \neq x\_j: \left\| x\_i - x\_j \right\|\_2 \le \epsilon\}$$
> >
> >Formally, this leads to a filtration of simplicial complexes $\{{\mathfrak{R}}\_{\epsilon\_0}(X\_{b\_m}), {\mathfrak{R}}\_{\epsilon\_1}(X\_{b\_m}), \ldots, {\mathfrak{R}}\_{\epsilon\_m}(X\_{b\_m}) \}$, with an ordered sequence of distance thresholds $0 = \epsilon\_0 < \epsilon\_1 < \ldots < \epsilon\_m$ (Figure 1). Persistent Homology (PH) tracks the evolution of topological features (e.g., connected components ($0$D), loops ($1$D), and voids ($2$D)) across different distances. Given the distance matrix of instances $A$ at each space, we consider the sorted distance values as the $\epsilon$ set. For each $\epsilon\_i \in \epsilon$ set, if a topological feature is born or disappears, $\epsilon\_i$ is recorded as a persistent edge, and the pair of data points whose interaction causes this event is recorded as the persistence pair $\pi\_i$. For example, let us consider $0$D topological features for two connected components of points, $\{x\_1,x\_2,x\_3\}$ and $\{x\_4,x\_5,x\_6\}$. Let $\epsilon\_2$ be the distance between $x\_1$ and $x\_6$, representing the smallest distance that connects these two components. In this case, $\epsilon\_2$ is the persistent edge, and $(x\_1,x\_6)$ forms the corresponding persistent pair.
>
> **With respect to the distance metric**: We use the Euclidean distance (or Norm2 distance) between two persistent edges. We clarified this by rewriting the objective function paragraph:
> >The topological signature of the bag in input space is the persistent pairings $\pi^{X\_{b\_m}}$ and their corresponding persistent edges in matrix $A^{X\_{b\_m}}$. Similarly, the topological signature of the bag in the latent space is $\pi^{Z\_{b\_{m}}}$ and $A^{Z\_{b\_{m}}}$. Our objective is to incorporate the topological signature of the input space as an inductive bias into the model, thereby enhancing its robustness and generalizable instance representation. Thus, we define a loss term to penalize the encoder $f\_{\theta}$ for any inconsistency in the bag's topological signature during projection from input to latent space. To this end, we utilize the topological loss proposed by  Moor et al. (2020b), which enables backpropagation through topological descriptors. This loss compares the persistent edges obtained from the input and latent spaces using the same persistent pairings and the $\ell_2$ norm. Specifically, we first compare the persistent edges in two input and latent spaces using the input persistence pairs, resulting in $L\_{X\_{b\_m}\rightarrow Z\_{b\_m}}$. Then, we compare the persistent edges in the two spaces using the latent persistence pairs, resulting in $L\_{Z\_{b\_m}\rightarrow X\_{b\_m}}$. We need to account for both losses because we want to enforce consistency between both mappings (data $\rightarrow$ latent and latent $\rightarrow$ data), while remaining invariant to the ordering of instances within each bag.

---

> ### Author Response · Authors · 2025-12-25
>
> **Requested Change 3 [findings of [1]]**: Thank you for bringing this important point to our attention. We have added the following paragraph to the discussion section:
> >**Instance Learnability.** To formalize the notion of instance learnability in weakly supervised MIL, we consider the standard MIL setting as follows. Each bag $B = \{x\_1,\dots,x\_n\} \subset \mathcal{X}$ contains instances with latent instance labels $y\_i \in \{0,1\}$ for a binary classification task. The bag label is generated by a monotone aggregation rule
> >
> >$$Y\_{b\_M} = \phi(y\_1,\dots,y\_n),$$
> >
> >such as the standard OR assumption in MIL. Changing any instance label from negative to positive cannot change a positive bag label to a negative one. This assumption holds for the aggregation functions used in this work. As our proposed method does not introduce any new aggregation function, we omit further demonstrations for the aggregations considered. We focus on the distinction between instance-level and bag-level operations, where $g\_\psi$ represents the bag-level operation, which jointly encompasses the aggregation function and the predictor.
> >
> >$$\hat{Y}\_{b\_M} = g\_\psi\big(\{ f\_\theta(x\_i) \}\_{i=1}^n \big).$$
> >
> >Following Jang & Kwon (2024), we define the bag-level risk as
> >
> >$$R\_{\mathrm{bag}} = \mathbb{E}\big[ loss(\hat{Y}, Y) \big]$$
> >
> >and the instance-level risk as
> >
> >$$R\_{\mathrm{inst}}(h) = \mathbb{E}\big[ loss(h(f(x)), y) \big],$$
> >
> >where $h(.)$ is an explicit instance predictor.
> >Considering $R\_{\mathrm{inst}}(h)$ as the true instance risk, there exists an empirical instance risk $\hat{R}\_{\mathrm{inst}}(h)$ trained on the finite training set of size $m$
> >
> >$$\hat{R}\_{\mathrm{inst}}(h) = \frac{1}{m} \sum\_{i=1}^{m}loss(h(f(x)), y).$$
> >
> >In general $R\_{\mathrm{inst}}(h) \neq \hat{R}\_{\mathrm{inst}}(h)$ due to finite training data and model capacity. The larger this gap becomes, the poorer the instance learnability is.
> >
> >In our framework, the topological loss acts exclusively on the instance encoder $f\_\theta$ and therefore influences only the instance-level risk $R\_{\mathrm{inst}}$. It does not modify the aggregation function or the bag-level predictor $g\_\psi$, and hence cannot directly reduce the bag-level risk $R\_{\mathrm{bag}}$. Instead, the topological inductive bias restricts the hypothesis space of the instance encoder, improving the learnability and robustness of instance-level representations. Such improvements at the instance level can indirectly benefit bag-level inference through the shared encoder, while preserving the standard MIL aggregation assumptions. Thus, our proposed approach satisfies the necessary conditions for instance learnability without introducing additional instance-level supervision or altering the bag-level decision rule.
>
> **Concern 1: Safety and Error Modes**: In the anemia experiments, we explicitly report Recall in Table 3, which is the False Negative rate and is critical in rare disease diagnosis. TG-MIL achieves a Recall higher than MIL baselines, while improving overall performance metrics, indicating that the proposed method does not increase missed diagnoses even under data scarcity. We now clarify this point in the manuscript by adding the following sentence to the anemia experiments:
> >Recall is reported to assess sensitivity to missed positive cases. All metrics are macro-weighted because of the dataset imbalance.
>
> **Concern 2: Dataset Bias and Generalizability**: We fully agree that generalizability to out-of-distribution data is an important consideration. However, at present, there is no publicly available dataset addressing the same clinical task, nor any dataset with a similar imaging modality or disease focus. Unfortunately, this makes external or cross-domain validation not possible at this stage.
>
> **Concern 3: Misinterpretation of Instance Predictions**: We understand the reviewer’s concern. To address this, we added a dedicated subsection in the Discussion section explicitly clarifying the intended clinical use of the proposed method.
> >**Broader Clinical Impact Statement.** Notably, in the current clinical setting, AI tools based on our method should be used strictly as decision support systems for clinicians, and not as standalone diagnostic oracles, due to the weakly supervised nature of MIL, potential misclassifications, data distribution shifts, and clinical liability considerations.

---

> > ### Comment · Reviewer_B9LH · 2026-01-20
> > **Not Recommend**
> >
> > While the authors addressed my initial concerns, the discussion with Reviewer KdtF highlights a serious concern regarding the depth of the contribution:
> >
> > Framing vs. Implementation: Since the method is restricted to 0D features, it effectively acts as a regularizer for MST edges. Labeling this as "Topological Guidance" while only using connectivity (0D) feels like an overstatement. The authors must justify why this complex TDA framework is superior to simpler graph-based distance constraints.

---

> > > ### Author Response · Authors · 2026-01-20
> > >
> > > We agree that for the distance-based filtrations we are using, 0D persistence is equivalent to single-linkage clustering and could be calculated with the same algorithms. However, we decided to use the language of persistent homology to make it clear that we have a stable summary, which can potentially support higher-dimensional topological features with other filtrations. In terms of computational complexity, our approach is not more complex than an MST calculation, but it has the added advantage that the resulting features serve as a quantitative summary. Notice that our formulation also accommodates "non-metric" and function-based filtrations, thus providing a lot of flexibility for extensions. We will adjust the wording in the manuscript carefully to reflect this fact. Would the reviewer be satisfied with this solution?

---

> ### Author Response · Authors · 2026-01-22
>
> To address any remaining concerns about the extendability of our framework to support higher-order topological features,  we calculated $L_{topo}$ for 1D (loops) and 2D (voids) topological features, as well as 0D (cc) on the Benchmark datasets (the lightest dataset). Accordingly, we updated Table 2. We hope these experiments will address the reviewer’s concerns in point 1.
>
> Added information to Table 2:
> >| Method                          | MUSK1      | MUSK2       | FOX            | TIGER        | ELEPHANT  |
> >|--------------------------------|-----------------|-----------------|-----------------|-----------------|------------------|
> >| TG-RGMIL dim:[0,1]      | 96.6 ± 4.8   | 96.7 ± 4.4   | 78.5 ± 5.0   | 95.3 ± 4.1   | 94.2 ± 5.1     |
> >| TG-RGMIL dim:[0,1,2]   | 98.2 ± 3.4   | 96.4 ± 3.5   | 79.2 ± 5.1   | 95.7 ± 4.2   | 96.3 ± 5.4     |

---

### Review · Reviewer_YjLR · 2025-11-29

**Summary Of Contributions:**

This paper focuses on Multiple Instance Learning with limited training data. To this end, the authors propose Topologically Guided Multiple Instance Learning (TG-MIL), a framework that introduces a topological inductive bias into the MIL training process. Specifically, the authors assume that the topological features of the bag should be preserved after embedding bag instances into the model’s latent space, ensuring that the essential topological properties of the data are maintained. Such inductive bias is introduced via a topological loss term $L_{topo}$ that penalizes discrepancies between the topological structures of the input space and the latent embedding space.

**Additional Comments:**

Major revision

**Audience:**

Yes

**Audience Explanation:**

Multiple Instance Learning (MIL) is a classic and widely adopted paradigm for weakly supervised learning. Even in the era of large models, MIL remains highly relevant, particularly in biomedical research. This paper introduces an interesting perspective, suggesting that preserving the topological structure of bags can enhance performance, especially in data-scarce scenarios. Therefore, I think the MIL-related theoretical insights presented in this paper will be of interest to a portion of the TMLR’s audience.

**Claims And Evidence:**

Yes

**Claims Explanation:**

Synthetic Datasets, five benchmark MIL datasets, and a dataset for anemia classification are included for evaluation. The experimental results generally prove the effectiveness of the proposed TG-MIL.

However, some experimental details are not clearly described. For example, regarding the benchmark experiments, it is not entirely clear whether TG-MIL and RGMIL utilized identical network structures particularly the pooling layers, at least based on the main text alone.

**Requested Changes:**

1. The experimental section should describe the experimental setup in greater detail, particularly within the main text. For instance, it is currently unclear whether TG-MIL maintained the same network architecture as RGMIL during the benchmark experiments. If the architectures differed, ablation studies on the benchmark datasets are missing and should be added.

2. It is recommended to include a hyperparameter sensitivity analysis, specifically examining the impact of the topological loss weight on model performance.

3. The notation in the Methods section requires more detailed explanation, particularly regarding symbols representing dimensions and quantities.

4. The computational cost of the topological loss is currently limited to theoretical analysis and lacks experimental support. I suggest adding empirical evidence in the revised version.

5. The authors introduce a real-world MIL application regarding anemia classification. To foster subsequent research in this field, I encourage the release of this dataset, provided that ethical and privacy constraints allow.

---

> ### Author Response · Authors · 2025-12-25
>
> **Point 1**: We appreciate the reviewer's suggestion. To ensure fairness in our comparisons, we employed the same network architecture as RGMIL in the benchmark experiments. We reimplemented RGMIL (without any topological inductive bias) and reported its performance in Table 2 as 'RGMIL'. We then added our topological inductive bias to the exact same network architecture and reported the results as TG-MIL. Thus, Table 2 in its current form actually contains the requested ablation, which is RGMIL with and without topological inductive bias.
> To clarify this point for readers in the revised manuscript, we now change the name of TG-MIL to TG-RGMIL, which incorporates topological inductive bias to the RGMIL model. Also, we highlight this point by adding the following text to section 5.2:
>
> >[...] For the aggregation function, we opted for regressor-guided pooling. In this way, the only modification to the network architecture of RGMIL in our setup (TG-RGMIL)  is integrating the topological signature calculator into instances’ input and latent space.
>
> **Point 2**: Following the reviewer’s suggestion, we have now included a coefficient search, examining the effect of the topological loss weight $\lambda$ on TG-MIL performance. For the benchmark dataset, we evaluated each dataset on five different $\lambda$ values (0.1, 0.05, 0.01, 0.005, 0.001). For anemia, we evaluated four pooling techniques across five different $\lambda$ values (0.05, 0.01, 0.005, 0.001, 0.0005).
> We updated the manuscript by adding Figures 4 and 5 and the following explanations to the Experiment section:
>
> Section Experiments, 5.2 MIL Benchmarks:
> >We selected the coefficient $\lambda$ heuristically by searching over a range of values compatible with the relative scales of the classification loss $L\_{\text{class}}$ and the topological loss $L\_{\text{topo}}$, as illustrated in Figure 4 (see Appendix Table 8).
>
> Section Experiments, 5.3 Anemia Classification:
> >We selected $\lambda = 0.005$, as it provides the strongest overall performance across pooling strategies. Notably, the performance is relatively stable across 2 orders of magnitude of $\lambda$ (see Figure 5 and Appendix Table 9).
>
> We now also provide the corresponding tables in the appendix:
>
> >TG-RGMIL Accuracy (%) Across $\lambda$ Values (mean ± std)
> >
> >| Dataset   | λ = 0.001        | λ = 0.005        | λ = 0.01         | λ = 0.05         | λ = 0.1          |
> >|-----------|------------------|------------------|------------------|------------------|------------------|
> >| MUSK1     | 95.2 ± 6.8       | 95.6 ± 5.8       | 91.2 ± 4.4       | 94.6 ± 7.8       | 92.8 ± 6.1       |
> >| MUSK2     | 93.6 ± 5.6       | 96.4 ± 5.0       | 97.8 ± 3.9       | 97.0 ± 4.2       | 99.6 ± 1.8       |
> >| FOX       | 73.6 ± 5.6       | 77.9 ± 10.8      | 72.5 ± 6.8       | 74.7 ± 5.4       | 76.4 ± 6.6       |
> >| TIGER     | 90.1 ± 4.7       | 88.8 ± 4.0       | 90.3 ± 5.0       | 96.1 ± 4.0       | 94.8 ± 5.0       |
> >| ELEPHANT  | 96.1 ± 3.8       | 90.2 ± 7.5       | 95.7 ± 2.7       | 94.1 ± 5.4       | 92.5 ± 3.4       |
>
>
>
> >TG-MIL Accuracy (%) Across $\lambda$ Values (mean ± std)
> >
> >| Pooling Method | λ = 0.0005        | λ = 0.001         | λ = 0.005         | λ = 0.01          | λ = 0.05          |
> >|----------------|------------------|-------------------|-------------------|-------------------|-------------------|
> >| Average        | 77.3 ± 4.5       | 82.3 ± 3.7        | 81.3 ± 2.5        | 77.5 ± 4.1        | 68.6 ± 18.2       |
> >| Max            | 65.1 ± 5.2       | 70.0 ± 2.8        | 71.4 ± 5.6        | 66.5 ± 8.0        | 64.8 ± 8.0        |
> >| Attention      | 75.9 ± 2.0       | 76.2 ± 2.8        | 77.8 ± 1.6        | 75.2 ± 5.0        | 69.2 ± 8.0        |
> >| Anomaly-aware  | 78.0 ± 5.0       | 77.7 ± 2.1        | 81.1 ± 2.2        | 82.2 ± 1.8        | 77.4 ± 3.4        |
>
>
> During this analysis, we noticed a typo in our previously reported results for TG-MIL on the anemia dataset with anomaly-aware pooling and corrected it in Table 3.

---

> ### Author Response · Authors · 2025-12-25
>
> **Point 3**: We gladly follow the reviewer’s suggestion and polished the text to avoid potential confusion between symbols denoting distances and dimensionality. For distance measurement in eq. 1 we replaced $d(a\_i, a\_j)$ with Norm-2 to prevent its conflict with d as dimensionality.
>
> **Point 4**: We thank the reviewer for raising this important point. In the revised manuscript, we have now added an empirical analysis of the computational cost of the proposed topological loss in the Experiments section:
>
> >**Practical Computational Cost.**
> We empirically evaluate the computational overhead introduced by the topological signature calculation using two complementary timing metrics: (i) the average training time per iteration, which reflects the end-to-end cost of one optimization step, including forward propagation, loss computation, and backpropagation, and (ii) the wall-clock forward time, defined as the real elapsed time during a single forward pass. The baseline MIL model requires $0.021$ seconds per iteration, whereas TG-MIL requires $0.077$ seconds per iteration, a $ 3.7\times$ increase in end-to-end training time per iteration. For a bag of size $n=200$, the wall-clock forward time increases from $0.15$ seconds for MIL to $0.35$ seconds for TG-MIL, a $2.3$ fold increase and a $\Delta t(200)=0.20$ seconds. For a bag of size $n=100$, the corresponding increase is from $0.12$ seconds to $0.18$ seconds ($1.5$ fold change, $ \Delta t(100)=0.059$ seconds). When the bag size doubles from $100$ to $200$, the additional forward-time overhead increases by a factor of approximately $3.5$, which is close to the factor of $4$ predicted by the quadratic complexity analysis. Since TG-MIL introduces no additional learnable parameters, these observations support the quadratic dominant term identified in the theoretical analysis.
>
> **Point 5**: Unfortunately, we do not have an ethics approval in place to publish this data. However, we acknowledge that accessing the data is possible upon request. We thus added “The anemia data can be shared upon request” to the acknowledgment section.

---

### Review · Reviewer_KdtF · 2025-12-21

**Summary Of Contributions:**

The paper proposes a method to improve MIL frameworks in data-scarce regimes. The authors introduce a topological loss term on both the input bag (image space) and the latent bag embedding. The loss forces the latent space topology (leveraging VR Filtration) to preserve the persistent pairings found in the input space. They evaluate this on synthetic MNIST/FashionMNIST bags, standard MIL benchmarks (MUSK, Fox, etc.), and a rare anemia dataset, claiming improvements in low-data settings.

**Audience:**

Yes

**Audience Explanation:**

Yes, there is an audience for this paper because data scarcity is a massive, real-world headache, especially in medical fields like rare disease diagnosis. Even though the math has holes, the core idea, using the "shape" or structure of data to help models learn from just a few examples, is an intuitive and promising direction. Researchers struggling with expensive or limited datasets will want to see if this "structural" approach can help them, regardless of the theoretical flaws.

**Broader Impact Concerns:**

No major ethical concerns that would require rejecting the paper, but there is a missing discussion on medical safety.

Since the paper focuses on diagnosing diseases (like anemia) using very small datasets, there is a risk that the model might not be reliable enough for real-world patient care. When AI is trained on very few examples ("data-scarce scenarios"), it is easy for the model to be biased or fail when it sees a patient who is slightly different from the small training group.

**Claims And Evidence:**

No

**Claims Explanation:**

Addressing data scarcity in MIL is a valid and important problem, particularly for medical applications like rare disease classification, where labeled bags are expensive. The method leveraging VR Filtration to describe the topology, which is also different from related graph-based methods, requires using a dataset prior to describing the topological relations. The authors included a section on CO2 emissions, which is good.

However, the submission fails to meet the TMLR criterion for claims supported by clear and convincing evidence. There is a substantial disconnect between the paper's broad theoretical claims regarding "topological inductive bias" and the limited empirical evidence provided by the actual implementation:

- The authors repeatedly claim to leverage "geometrical-topological" information, explicitly citing features like "loops (1D), higher-dimensional voids (2D)"  to motivate their method. However, in the implementation details (Section 4, Complexity), they admit they "only capture 0-dimensional topological features" (connected components) to maintain computational efficiency. If I understand correctly, (0-dimensional) persistence is mathematically equivalent to analyzing the edge lengths of a single-linkage clustering. It essentially acts as a pairwise distance regularizer that enforces connectivity.

- The authors frame the treatment of an MIL bag as a "point cloud" as a core contribution and insight. This ignores foundational literature in Geometric Deep Learning (e.g., PointNet, Deep Sets) which established the equivalence between processing point clouds and permutation-invariant sets years ago. Presenting a standard problem definition as a novel insight weakens the credibility of the submission's contributions.

- Fig. 2 is also a little bit vague. Why not use ABMIL as the baseline?

**Requested Changes:**

- Again, as I mentioned earlier, if I understand correctly for the 0-dimension implementation. The authors need to rephrase the claim. i, e. The paper repeatedly claims to leverage "geometrical-topological" information, explicitly mentioning "loops" and "higher-dimensional voids" in the Introduction. However, Section 4 admits the method is restricted to 0-dimensional features ($H_0$) due to complexity constraints.

- Rewrite the analysis of Figure 2 to remove the claim that the "circle" shape is a direct result of the topological loss.

- Cite Foundational Geometric Deep Learning Literature The paper frames "treating a bag as a point cloud" as a novel perspective, e.g,. Deep Sets.

---

> ### Author Response · Authors · 2025-12-25
>
> **Point 1: [The authors repeatedly claim to …]**: We confirm that all reported results are obtained using 0-dimensional topological features (connected components), as explicitly stated in the manuscript. However, this does not reduce the topological loss to a simple pairwise distance regularization, such as an L2 loss. Instead, it captures critical pairs and their corresponding distances that govern the birth and merging of connected components. As a result, the proposed loss preserves the 0-dimensional topological structure of bags and introduces a topological inductive bias at the level of instance connectivity.
> To clarify this distinction, we revised and expanded the Method section to explicitly describe persistent pairs and edges, and we added an illustrative example to make the loss behavior more transparent.
> >We calculate the simplicial complex of Vietoris-Rips (${\mathfrak{R}}\_{\epsilon}(X)$) at each distance $\epsilon$, which yields all subsets of $X$ such that the pairwise distance between points is less than or equal to $\epsilon$.
> >
> >$${\mathfrak{R}}\_{\epsilon}(X\_{b\_m}):= \{x\_i| \exists x\_j \in X , x\_i \neq x\_j: \left\| x\_i - x\_j \right\|\_2 \le \epsilon\}$$
> >
> >Formally, this leads to a filtration of simplicial complexes $\{{\mathfrak{R}}\_{\epsilon\_0}(X\_{b\_m}), {\mathfrak{R}}\_{\epsilon\_1}(X\_{b\_m}), \ldots, {\mathfrak{R}}\_{\epsilon\_m}(X\_{b\_m}) \}$, with an ordered sequence of distance thresholds $0 = \epsilon\_0 < \epsilon\_1 < \ldots < \epsilon\_m$ (Figure 1). Persistent Homology (PH) tracks the evolution of topological features (e.g., connected components ($0$D), loops ($1$D), and voids ($2$D)) across different distances. Given the distance matrix of instances $A$ at each space, we consider the sorted distance values as the $\epsilon$ set. For each $\epsilon\_i \in \epsilon$ set, if a topological feature is born or disappears, $\epsilon\_i$ is recorded as a persistent edge, and the pair of data points whose interaction causes this event is recorded as the persistence pair $\pi\_i$. For example, considering $0$D topological features, suppose we have two connected components of points, $\{x\_1,x\_2,x\_3\}$ and $\{x\_4,x\_5,x\_6\}$. Let $\epsilon\_2$ be the distance between $x\_1$ and $x\_6$, representing the smallest distance that connects these two components. In this case, $\epsilon\_2$ is the persistent edge, and $(x\_1,x\_6)$ forms the corresponding persistent pair.
>
> **Point 2: [The authors frame the treatment …]**: We believe this comment stems from a misinterpretation of our claims of contributions.
> We would like to clarify that we do not claim that treating a bag as a point cloud is our novel contribution. By definition, a MIL bag consists of a set of instances and therefore naturally exhibits a point-cloud structure in the underlying feature space. Thus, viewing a bag as a point cloud is inherent to the MIL formulation and not an insight we attribute to our work.
> Our contribution instead lies in highlighting the beneficial role of topological analysis of this point-cloud structure in the data-space as an inductive bias for MIL in data-scarce scenarios. To the best of our knowledge, prior works did not provide such a contribution before.
>
> **Point 3: [Figure 2]**: We would like to point out that Figure 2 is purely illustrative and therefore included in the Methods section rather than as an experiment. We opted to use regressor-guided pooling in this figure to remain consistent with (i) the best-performing pooling approach on the synthetic dataset and (ii) our experimental choices on the MIL benchmark datasets. Nevertheless, our proposed method is not tied to a specific pooling mechanism and can be applied to alternative aggregation schemes, including attention-based pooling, as well as max or average pooling. The instance-level analysis of the anemia experiments is a similar empirical illustration of an attention-based pooling. There, we provide additional visualizations in the form of distance matrices that serve as an implicit indication that incorporating a topological inductive bias encourages instance representations to remain overall aligned with the underlying data distribution in the Euclidean metric space.

---

> > ### Comment · Reviewer_KdtF · 2026-01-05
> >
> > I thank the authors for their detailed response.
> >
> > Regarding Point 1 (0D Topological Features):
> > I understand and agree that the 0D persistent homology loss operates on "critical pairs" (edges in the Minimum Spanning Tree) rather than all pairwise distances. This is a valid distinction from simple L2 regularization.
> > However, the core issue remains one of framing. The Introduction and method sections explicitly motivate the method using concepts like "loops (1D)" and "voids (2D)" (e.g., citing general TDA properties). Since the implementation is strictly limited to 0D features (connectivity/clustering), referencing higher-order topological features in the motivation creates a misleading expectation that the model captures complex manifold shapes, which $H_0$ theoretically cannot do.
> >
> >
> > Regarding Point 2 (Bag as Point Cloud & Missing Citations): I note your clarification that you view the "point cloud structure" as inherent to MIL. However, your response failed to address my specific request to cite the foundational literature. In my initial review, I explicitly pointed out that works like Deep Sets (Zaheer et al., 2017)  established the theoretical framework for learning on permutation-invariant sets to prevent any future "misinterpretation" by readers.
> >
> > [1] Deep Sets, https://arxiv.org/abs/1703.06114

---

> ### Author Response · Authors · 2026-01-08
>
> We thank the reviewer for clarifying the mentioned points and adapting our manuscript accordingly
>
> **Point 1**: While the proposed loss, in its current formulation, theoretically supports the inclusion of higher-order topological features (and our released code also provides such an extension), we have had to restrict our experiments to 0D features due to computational complexity considerations.
>
> However, we completely understand and agree with the reviewers’ concerns about ensuring that readers are not misled regarding the use of higher-order topological features in the current manuscript. To prevent potential misunderstanding, we have explicitly clarified this point in the Methods section of the revised manuscript.
>
> In the Introduction section, we revised the text to be:
> >“We demonstrate that maintaining (even only) the connectivity-based topological bias (captured via $0$D persistent homology) inherent in a bag's data distribution improves the performance of MIL classifiers, even with varying amounts of data.”
>
> In the Method section, we added:
> >“Consistent with previous  work (Moor et al., 2020), we restrict most of our experiments to $0$D topological features, as incorporating higher-dimensional topological features substantially increases computational cost.“
>
> **Point 2**: We have clarified the relevant point in this paper and revised the corresponding sentence in the introduction to prevent possible misinterpretation by readers.
>
> “By treating each bag as a point cloud in a high-dimensional data space, we can define an inductive bias based on the topological features of the bag. ”
>
> is rewritten as:
>
> >“We can consider each bag as a permutation-invariant point cloud in a high-dimensional data space, consistent with Deep Sets and Pointnet (Zaheer et al., 2017; Qi et al., 2017), and define an inductive bias based on the topological features of the bag.”

---

> ### Author Response · Authors · 2026-01-22
> **Demonstration of higher-order topological features support**
>
> As a demonstration of the extendability of our framework to cover higher-order topological features, on the Benchmark datasets (the lightest dataset), we calculated $L_{topo}$ for 1D (loops) and 2D (voids) topological features, as well as 0D (connected components), and updated Table 2, its caption, and the corresponding explanation in the Result section accordingly. We hope these experiments will address the reviewer’s concerns in point 1.
>
> Added information to Table 2:
> >| Method                          | MUSK1      | MUSK2       | FOX            | TIGER        | ELEPHANT  |
> >|----------------------------|-----------------|-----------------|-----------------|-----------------|------------------|
> >| TG-RGMIL dim:[0,1]      | 96.6 ± 4.8   | 96.7 ± 4.4   | 78.5 ± 5.0   | 95.3 ± 4.1   | 94.2 ± 5.1     |
> >| TG-RGMIL dim:[0,1,2]   | 98.2 ± 3.4   | 96.4 ± 3.5   | 79.2 ± 5.1   | 95.7 ± 4.2   | 96.3 ± 5.4     |
>
> Added text to caption:
> >“The last three rows correspond to TG-MIL results obtained by incorporating topological features of increasing homological dimensions. The configurations [0]D, [0,1]D, and [0,1,2]D progressively include 0-, 0 and 1-, and 0, 1, and 2-dimensional topological features, respectively.”
>
> Modified text in 5.2 MIL Benchmarks, Results:
> >“TG-RGMIL consistently outperforms non-topological approaches in all datasets. Incorporating higher-dimensional topological features generally improves performance across most cases (MUSK1, FOX, ELEPHANT datasets). For MUSK2 and TIGER, TG-MIL, considering higher-order features, remains competitive with TG-MIL using only 0D topological features, while still outperforming state-of-the-art methods. In these two datasets, the underlying instances' topology is likely dominated by connectivity; thus, higher-dimensional topological features do not provide additional class-discriminative cues. “

---

### Review · Reviewer_wSEY · 2025-12-22

**Summary Of Contributions:**

The authors introduce Topologically Guided Multiple Instance Learning (TGMIL), a novel method to incorporate topological inductive biases into deep MIL models. Specifically, the authors utilize a topology-preserving constraint to ensure that the topology of the instances within a bag are maintained when mapped from the input space to the MIL latent space. The proposed framework is evaluated across synthetic datasets, standard MIL benchmarks, and a real-world clinical application involving rare anemia classification.

**Strengths.**
- Clarity and presentation. The paper is well written and is easy to follow.
- Novelty. This work represents the first attempt to incorporate topological data analysis (TDA) into the MIL paradigm.
- Flexibility. A major advantage of TGMIL is its modularity; the topological loss can be integrated into any existing deep MIL architecture.
- Convincing experimental section. Although the methods considered for comparison are limited (see next section), the experimental results look convincing: the proposed methodology improves performance, specially when training data is limited.

**Weaknesses / Suggestions.**
- Accessibility of TDA concepts. The paper assumes substantial prior knowledge of Topological Data Analysis (TDA), which may limit accessibility for a broader audience. Although the authors reference the loss formulation introduced in prior work, the presentation would be strengthened by including a concise algorithmic description or an explanatory paragraph detailing the step-by-step computation of persistence diagrams.
- Meaning of 0-dimensional topological features. Due to efficiency constraints, the topological loss is restricted to 0-dimensional topological features. For someone who is not well-versed in TDA, it is not clear what information these features capture. It would be helpful if the authors could clarify this point.
- Omission of modern MIL architectures. Recent MIL methods are completely ignored. It is well-known that accounting for (local or global) interactions between instances can lead to significant performance gains in MIL. This has been done through Transformers [1], Graph Neural Networks (GNNs) [2], or the combination of both [3,4]. Since the proposed methodology explicitly incorporates information about interactions among instances within a bag, a discussion of these related approaches (and their relationship to the proposed method) would be highly relevant.
- Extension of the experimental section. In the experimental section, performance is evaluated only by applying TGMIL on top of relatively simple aggregation mechanisms. As noted above, the proposed framework could naturally be combined with more expressive aggregation models. In particular, it would be interesting to see how TGMIL influences the learned representation space of a Transformer-based MIL model. Note that the current results suggest that TGMIL consistently improves performance across various aggregation mechanisms, and thus it is expected that this result will carry over to more powerful architectures. If considering these architectures for comparison does not make sense, it would be useful to explain why.

**Other comments.**
- In the anemia classification task, instance-level analysis is done qualitatively (Figure 4). Can this be evaluated quantitatively? If instance-level labels are available, classification performance metrics could be computed using the model’s anomaly score.
- Figures 7 and 8 in the Appendix aim to illustrate how TGMIL mitigates overfitting through learning curves. However, the loss scale differs across rows. It is also unclear which loss is being reported. If the curves correspond solely to the cross-entropy loss, I don’t see an obvious justification for using different scales. As presented, these plots are not comparable.

**References.**

[1] Shao, Zhuchen, et al. "Transmil: Transformer based correlated multiple instance learning for whole slide image classification." Advances in neural information processing systems 34 (2021): 2136-2147.

[2] Tu, Ming, et al. "Multiple instance learning with graph neural networks." arXiv preprint arXiv:1906.04881 (2019).

[3] Castro-Macías, Francisco M., et al. "Sm: enhanced localization in Multiple Instance Learning for medical imaging classification." Advances in Neural Information Processing Systems 37 (2024): 77494-77524.

[4] Fourkioti, Olga, et al. "CAMIL: Context-aware multiple instance learning for cancer detection and subtyping in whole slide images." arXiv preprint arXiv:2305.05314 (2023).

**Audience:**

Yes

**Audience Explanation:**

This is the first work exploring the use of topological information in MIL. It is applied to an important medical-imaging problem. It shows promising results and this could potentially advance the research frontier in MIL.

**Broader Impact Concerns:**

I don't identify any major ethical concern.

**Claims And Evidence:**

No

**Claims Explanation:**

The authors claim that “TG-MIL outperforms the state-of-the-art on MIL benchmarks and rare anemia classification”. As explained before, the comparison against several SOTA baselines is missing. If the comparison does not make sense, then the authors should explain why.

**Requested Changes:**

I believe that this is a good paper that should be considered for acceptance in TMLR. I suggest that the authors take into accounts the recommendations listed in the “Weaknesses / Suggestions” and “Other comments” sections before.

---

> ### Author Response · Authors · 2025-12-25
>
> **Points 1 & 2: Accessibility of TDA concepts & Meaning of 0-dimensional topological features.** We thank the reviewer for bringing these points to our attention. We revised and expanded the Method section to explicitly describe persistent pairs and edges, making the loss behavior more clear. Also, we added an example of persistent pairs and edges in one step of the persistent homology algorithm for 0-dimensional topological features:
>
> >We calculate the simplicial complex of Vietoris-Rips (${\mathfrak{R}}\_{\epsilon}(X)$) at each distance $\epsilon$, which yields all subsets of $X$ such that the pairwise distance between points is less than or equal to $\epsilon$.
> >
> >$$ {\mathfrak{R}}\_{\epsilon}(X\_{b\_m}):= \{x\_i| \exists x\_j \in X , x\_i \neq x\_j: \left\| x\_i - x\_j \right\|\_2 \le \epsilon\}$$
> >
> >Formally, this leads to a filtration of simplicial complexes $\{{\mathfrak{R}}\_{\epsilon\_0}(X\_{b\_m}), {\mathfrak{R}}\_{\epsilon\_1}(X\_{b\_m}), \ldots, {\mathfrak{R}}\_{\epsilon\_m}(X\_{b\_m}) \}$, with an ordered sequence of distance thresholds $0 = \epsilon\_0 < \epsilon\_1 < \ldots < \epsilon\_m$ (Figure 1). Persistent Homology (PH) tracks the evolution of topological features (e.g., connected components ($0$D), loops ($1$D), and voids ($2$D)) across different distances. Given the distance matrix of instances $A$ at each space, we consider the sorted distance values as the $\epsilon$ set. For each $\epsilon\_i \in \epsilon$ set, if a topological feature is born or disappears, $\epsilon\_i$ is recorded as a persistent edge, and the pair of data points whose interaction causes this event is recorded as the persistence pair $\pi\_i$. For example, considering $0$D topological features, suppose we have two connected components of points, $\{x\_1,x\_2,x\_3\}$ and $\{x\_4,x\_5,x\_6\}$. Let $\epsilon\_2$ be the distance between $x\_1$ and $x\_6$, representing the smallest distance that connects these two components. In this case, $\epsilon\_2$ is the persistent edge, and $(x\_1,x\_6)$ forms the corresponding persistent pair.

---

> ### Author Response · Authors · 2025-12-25
>
> **Points 3 & 4: Omission of modern MIL architectures & Extension of the experimental section.** While we also find it interesting to study the effect of TG-MIL when combined with other aggregation mechanisms, executing a fair comparison with these approaches is challenging due to differences in problem definition. As discussed in the Related Work section, recent MIL methods can be broadly categorized into: (i) approaches that rely on transfer learning for feature extraction and focus on training aggregation mechanisms, and (ii) end-to-end approaches that jointly optimize instance representations and bag aggregation.
> The suggested works [1,3,4] fall into the first category. They rely on frozen, pre-trained instance encoders and are designed to model spatial dependencies between instances, which are intrinsic to their datasets (e.g., whole-slide imaging). In contrast, TG-MIL targets the improvement of instance representation learning in an end-to-end MIL framework and is evaluated on datasets with spatially independent instances. Directly applying these methods would therefore require architectural modifications to match our problem setting, and would not result in a fair comparison.
> To clarify why these approaches are excluded from our experimental comparisons, we rewrote the first paragraph of the Related Work as follows:
>
> >“Recent advances in MIL can be categorized into two main methods and use cases:
> i) Methods using transfer learning for feature extraction and only focusing on training aggregation mechanisms (Shao et al., 2021; Zhang et al., 2022; Liu et al., 2023; Tang et al., 2023; Fourkioti et al., 2023; Castro-Macías et al., 2024). These approaches are designed for histopathology applications and have not been evaluated on basic MIL problem settings. Bags typically consist of thousands of spatially correlated patch instances extracted from a single whole-slide image. Due to architectural and computational constraints, instance representations are kept fixed using frozen pre-trained encoders, and end-to-end training is not supported. In contrast, our work focuses on improving instance representation learning in end-to-end MIL settings with spatially independent instances and limited training data. Adapting the above methods would require removing certain components (e.g., positional embeddings in TransMIL), reducing the complexity of the aggregation model, and considering a proper learnable instance encoder. These changes would substantially alter the original architectures and make a fair comparison non-trivial. Therefore, we exclude these approaches from aggregation comparisons, while noting that our proposed framework is, in principle, compatible with such aggregation mechanisms. [...]”
>
> Among the suggested methods, only GNN-MIL [2] reports results on MIL benchmark datasets. Following the reviewer's comment, we have now added GNN-MIL [2] to Table 2 for completeness. We would like to note that Table 2 already includes more recent end-to-end MIL approaches, several of which outperform [2].

---

> ### Author Response · Authors · 2025-12-25
>
> **Comment 1 [instance-level analysis of anemia]**: Unfortunately, instance-level labels are not available for the anemia dataset; only patient-level diagnoses are provided. As a result, quantitative evaluation of instance-level predictions is not possible at this point.
>
> **Comment 2 [loss scale]**: We thank the reviewer for raising this point. Figures 7 and 8 report the final training objective: for RGMIL, this corresponds to the bag-level cross-entropy loss, while for TG-RGMIL, it is the total loss (cross-entropy + topological loss). Since the topological loss has a larger scale, the total loss magnitude is higher for TG-MIL, particularly at the beginning of training. The coefficient $\lambda$ controls the relative contribution of the topological term. Despite the different loss scales, TG-MIL consistently converges to a lower final total loss. This indicates that the cross-entropy component also converges to a lower value in TG-MIL compared to MIL, making the two approaches comparable at convergence. Now we added the following text to the appendix to clarify this:
> >For RGMIL, the loss corresponds to the bag-level cross-entropy loss, while for TG-RGMIL, it is the total loss, defined as the sum of the cross-entropy and topological losses. Since the topological loss operates at a larger scale, the total loss magnitude for TG-RGMIL is higher, particularly during the early stages of training. Despite the different loss scales, TG-RGMIL consistently converges to a lower final total loss, indicating that the cross-entropy component also converges to a lower value compared to RGMIL, making the two approaches comparable at convergence.

---

> > ### Comment · Reviewer_wSEY · 2026-01-05
> >
> > I would like to thank the authors for their detailed response and for the effort they have made to improve the paper. I would like to raise two minor points regarding their rebuttal, which I mention here for completeness.
> >
> > Points 3 & 4: Omission of modern MIL architectures and extension of the experimental section. I understand the authors’ rationale for not including these architectures in the experiments. Nevertheless, I believe that an analysis of how the proposed topological loss affects the instance representations learned by a Transformer-based model would further strengthen the paper and add complementary insight.
> >
> > Comment 2 (loss scale). While the additional explanation helps clarify the issue, I still believe that reporting the cross-entropy component separately would be beneficial, as it would more clearly demonstrate that it attains a lower value.

---

> ### Author Response · Authors · 2026-01-08
>
> We thank the reviewer for the follow-up and are pleased that our previous clarifications were helpful.
>
> **Points 3&4**: We agree with the reviewer that analyzing the effect of the proposed topological loss within end-to-end Transformer-based MIL models could provide additional insight. However, performing such an analysis in a principled manner is non-trivial and beyond the scope of the current work, as it introduces many additional considerations related to architectural design.
>
> In addition to the substantial modifications mentioned earlier, which would compromise the fairness of the experiments, there are further concerns regarding the application of the proposed topological inductive bias to Transformer-based aggregation mechanisms. In Transformer-based MIL architectures, the aggregation mechanism consists of multiple self-attention layers, each transforming the instance representations to a new space. As a result, there is no single, well-defined instance embedding space. A fair analysis would therefore require covering different design choices (with theoretical support) for where the topological loss should be applied:  before the aggregation, introducing a source instance representation (which contradicts the contribution of our work for harnessing scarce data), or in all self-attention layers, which directly restricts the transformer's concepts and necessitates careful consideration of its definition of constrains. Moreover, applying the topological loss across multiple attention layers would introduce non-trivial computational complexity in MIL, particularly for large bag sizes. We believe these constraints fall outside the scope of this analysis, although they represent interesting directions for future research.
>
> On the other hand, the models considered in this paper allow a clear separation between instance representation learning and aggregation (without such overlap of theoretical concepts), enabling a straightforward study of the proposed inductive bias.
>
> **Comment 2**: We thank the reviewer for this suggestion. To further clarify the effect of the proposed regularization, we have updated Figure 9 in the appendix to separately report the learning curves of the total loss ($L_{totoal}$), the MIL cross-entropy loss ($L_{class}$), and the topological loss ($L_{topo}$). This additional visualization explicitly shows that the cross-entropy component attains a lower value during training.

---

### Decision · Action_Editor_A8bb · 2026-01-27

**Recommendation:** Accept as is

**Audience:**

Yes

**Audience Explanation:**

The use of topological information on multiple instance learning, and the medical-imaging problems make this paper interesting to the TMLR audience.  Reviewer B9LH had mixed concerns about how appropriate the paper is to TMLR audience.  While the original review mentions that the paper is interesting to the audience the recommendation says "no", but is not supported by the comments.

**Claims And Evidence:**

Yes

**Claims Explanation:**

The majority of the reviewers agree that the claims of the paper are supported.  In particular, the multiple instance learning framework that is supported by the topological inductive bias.  Reviewer KdtF had concerns about the unsupported claims and evidence, which were also picked by reviewer B9LH. In my opinion, the authors replies cover the raised concerns.